# Evaluation of a scheme to identify risks for tail biting in pigs

Roberta Maria D'Alessio[1,2], Conor G. Mc Aloon[2], Carla Correia-Gomes[3], Alison Hanlon[2], Keelin O'Driscoll[1]*

1 Pig Development Department, Animal & Grassland Research & Innovation Centre, Teagasc Moorepark, Fermoy, Co. Cork, Ireland, 2 UCD Veterinary Sciences Centre, University College Dublin, Dublin, Ireland, 3 Animal Health Ireland, Carrick on Shannon, Co. Leitrim, Ireland

* keelin.odriscoll@teagasc.ie

**Data Availability Statement:** Data cannot be shared publicly because of GDPR considerations regarding farm audits. However, a request for farm audit data can be submitted to Animal Health Ireland (AHI) at ahicentral@animalhealthireland.ie

## Abstract

The study aimed to assess the effectiveness of a tail-biting risk assessment scheme. The scheme consisted of trained private veterinary practitioners (assessors) applying a risk assessment tool on commercial pig farms to six pens per farm. The assessment tool included animal and non-animal-based observations which were used to determine the perceived risk of tail biting for each pen. For this study 27 farms were assessed, and a subsequent batch of pigs from each farm underwent post-mortem tail lesion scoring at the abattoir. The assessments revealed that a high percentage of pens had fully slatted flooring (92%) and mixed-sex populations (84%), with a significant proportion of pens containing pigs which were all tail docked (92%). Most pens (86%) did not allow all pigs simultaneous access to feeders. Enrichment was present in 88% of the pens, but most (46%) were supplied with only one item, and only 15% offering multiple enrichment types. The study found no significant associations between the risk of tail biting and visible injuries, dirty flanks, or tucked tails, as assessed by the assessors (P > 0.05). Similarly, the risk of tail biting reported per pen was not associated with aggressive, damaging, or exploratory behaviours (P > 0.05). At the abattoir, 96% of pigs' tails exhibited minor skin damage, with only 4% showing moderate to severe damage. Furthermore, no links were found between the scores obtained during slaughter and the risk of tail biting, as reported by the assessors (P < 0.05). Although the tool was useful in identifying several improvements that could be made at farm level in areas such as stocking density, enrichment provision and reducing tail docking, overall the results underscored the need for improved training of assessors, and the challenge of associating management practices and animal based measures with tail-biting risk.

## 1. Introduction

Although European Directive 120/2008 prohibits routine tail docking on pig farms, this procedure is still commonly used in most European countries [1]. It is carried out because it reduces the frequency of tail biting, a significant health and welfare problem in pig production. Nevertheless, it does not entirely eliminate the problem, especially if husbandry deficiencies remain

AHI data are stored for at least 10 years for cross-sectional data and more than 10 years if part of a programme. Data collected by Teagasc researchers from the abattoir are available upon request by contacting Dr Keelin O'Driscoll (Teagasc Principal Investigator; keelin.odriscoll@teagasc.ie), Dr Edgar Manzanilla (Teagasc Head of Pig Development Department; Edgar.GarciaManzanilla@teagasc.ie) or Dr John Hyland (Teagasc Data Value Platform Manager; John.Hyland@teagasc.ie). These data will be archived with a metadata record on the Teagasc Data Value Platform, a soon to be operational institutional data repository. The platform is currently being trailed and the dataset will be stored on the platform once it is fully operational from May 2024.

**Funding:** The project was funded by Teagasc, the Irish Agricultural and Food Development Authority. Project no. 0842.

**Competing interests:** NO authors have competing interests Enter: The authors have declared that no competing interests exist.

unresolved [2]. In fact, data from farms and abattoirs suggest that 30% to 70% of farms in various European countries have some degree of tail-biting problem despite tail docking being prevalent [2]. Reports on tail lesion prevalence at the abattoir vary greatly, with EFSA estimating an overall prevalence across Europe of 10% in 2007 [2]. However, research in Irish abattoirs has reported much higher prevalence, from 20% in 2016, 30.8% in 2015 [3, 4] and even up to 72.5% in 2014 [5]. This variation can be caused by not only differences in the number of lesions present, but by the severity of lesions included in the assessment system, and different data collection points at the slaughter line. Consequently, meat inspection results and scientific studies may differ considerably [6].

In 2016, having noted the unsatisfactory implementation and enforcement of the Directive, the European Commission published European Recommendation 336 of 2016 'on the application of Council Directive 2008/120/EC laying down the minimum standards for the protection of pigs as regard measures to reduce the need for tail-docking' [7]. The recommendation required each European Union (EU) Member State to ensure farmers carry out a tail-biting risk assessment, and that corrective measures for each identified risk factor are recorded [7]. However, identifying risk factors can be challenging, and it is not only individual risk factors, but their combination, that can result in sporadic and unexpected tail biting outbreaks [8]. Causes of tail biting may differ between farms, or by production stage, and change within farms over time. As a result, farm-specific recommendations are necessary to reduce the risk of tail biting [9]. Using a structured risk assessment tool is a potentially effective option to provide an overview of the farm situation and support the farmer in implementing strategies to reduce the risk of tail biting. Through this, the producer, farm advisers, and veterinary practitioners, can all then play a role in identifying and implementing the most suitable interventions based on scientific knowledge, and their practical experience of the farm [10].

Recently, Dippel et al. [11] compiled a list of tail-biting risk factor assessment protocols developed in several European countries. However, only two of these have been scientifically validated, and the results published. Among these, the Husbandry Advisory Tool (HAT) was the first to be developed and tested by Taylor et al. [12], for UK pig production systems. An abbreviated version has been made available online, now termed the 'WebHAT' (https://webhat.ahdb.org.uk/). It is based on the concept of Animal Health and Welfare Planning (AHWP), a series of principles whereby risk factors and outcomes present on farm are evaluated, with subsequent planning of implementation of interventions and regular reviewing [9, 13]. This tool is divided into eight categories (namely environment, enrichment, feed/drink, health per pen, group variation in tail length, stocking density, tail biting history and transport events), with a total of 83 recognised risk factors included, as well as closed questions concerning the type of farm.

This study aimed to assess the effectiveness of an adapted version of the WebHAT tail-biting risk assessment tool in estimating the risk of tail biting on farm, as part of a national risk assessment scheme. We hypothesised that the level of risk determined by the tool would be reflected in tail lesion scores recorded at the slaughterhouse.

## 2. Material and methods

The study was carried out between February 2020 and May 2022. It was prolonged as a consequence of the COVID-19 outbreak and biosecurity measures, which resulted in interruptions to slaughterhouse access. As the experimental work carried out by the research team involved post-mortem carcase inspection at the slaughterhouse, no ethical approval was needed.

## 2.1 Development of the tool

Prior to the start of this study a team of pig welfare experts (including the last author of this paper) reviewed the questions in the WebHAT tool, and selected those which were relevant to Irish commercial pig production systems (e.g. all questions regarding outdoor farms, or farms where pigs were bedded on straw, were removed). The remaining questions were consolidated into a short assessment protocol, designed to take a maximum of 15 minutes to complete per pen by a trained assessor (the complete assessment is available in S1 File). Assessors were all Private Veterinary Practitioners that specialised in pig management and were the farm veterinarian.

The assessor was tasked with appraising six pens on each farm, distributed across distinct production stage stages. This encompassed two pens during the first weaning stage (from the day of weaning to approximately four weeks post-weaning), followed by an assessment of two pens at the second weaning stage (approximately four weeks post-weaning until moving to the finisher stage), and finally, an evaluation of two pens at the finisher stage. Within farms featuring a singular weaner stage, assessors were instructed to assess two pens of pigs weaned one to two weeks post weaning, two pens of pigs near the end of the weaner stage, and two pens in the finisher stage. In addition, on finishing-only farms, assessors were required to select pens at various stages of the fattening process. The evaluation of all designated pens across all stages was required to be completed on the same day.

The first part of the assessment comprised a section regarding pig housing and management. This included measurement of pen length and width, estimation of the proportion of solid flooring (0%, 1–25%, 26–50%, 51–75% or 76–100%), the sex of the pigs per pen (male, female or mixed sex), the final weight achieved in the pen (based upon an estimate provided by the producer), tail length of the pigs (docked, undocked or mixed length), whether pigs can all feed at the same time, the number of drinkers present, and whether the assessor considered the vaccination protocol to be appropriate. Next, the assessor was asked to detail the type and amount of environmental enrichment present in the pen. Enrichment type was classified according to the criteria described in the Commission Staff Working Document 2016 (i.e. optimal, sub-optimal or marginal) [14]. For sub-optimal and marginal items, the number provided in the pen were counted, and for optimal, presence or absence noted. Other than the final group weight in the pen, which could be considered a continuous variable, the answers to this section were either categorical (i.e. answers in tick boxes) or else count data (the enrichment items).

Following this the assessment involved the evaluation of physical and behavioural animal based welfare indicators. The assessor stood by the side of the pen, counted the pigs, and then counted the number of pigs affected by the following physical conditions: injured tails, injured or imperfect ears, flank lesions (circular), aggression lesions (straight). All conditions were evaluated from outside the pen, and severity was not considered, only presence or absence on each pig. They also counted the number of pigs with dirty flanks and tucked tails. Finally, they carried out all occurrence behaviour observations for 5 minutes, using an ethogram of the following behaviours: tail biting, ear biting, damaging biting of other parts of the body, investigation of fixture and fittings, investigation of enrichment material, and aggressive biting. Thus all of the animal based measures produced continuous data, as they required the assessor to count either the number of physical conditions, or performances of the behaviours in the ethogram.

At this stage the assessor was asked to state whether or not they considered there was an overall risk of tail biting occurring in the pen (Yes/No), using the results of the assessment as a guide. Following this, assessors were asked to consider in more detail the level of tail biting

**Table 1. The exact wording of the risk categories included in the risk assessment tool.**

| Risk Category |
| --- |
| **Environmental enrichment** provision represents no risk for tail biting |
| There is adequate **thermal comfort** and air quality for these pigs |
| The **health** of these pigs provides no risk of tail biting |
| **Competition** issues for the pigs in this pen do not give rise to risks for tail biting |
| The **pen design** and use for these pigs does not present risk for tail biting |
| **Feeding processes** for these pigs do not contribute to risks for tail biting for these pigs |

risk in relation to six risk categories, selected because these areas are outlined in Commission Recommendation (EU) 2016/336 [15], and again using their judgement based upon the outcome of the assessment. These six risk categories are: (a) the enrichment materials provided; (b) cleanliness; (c) thermal comfort and air quality; (d) health status; (e) competition for food and space; (f) diet. The exact wording of each risk category can be seen in Table 1.

The levels of risk that the assessors were asked to assign to each of these risk categories is described in Table 2. The definitions in Table 2 are exactly as described in the risk assessment tool. As further explanation, a level of 0 meant that the assessor did not observe any risk at all. Level 1 indicates that there is no risk of tail biting for that specific risk category (i.e. the statement, which implies no risk, is correct), and Level 2 implies that although risks may be present, the assessor was not able to identify specifically what they are. Levels 3 and 4 differ in that at Level 3 the assessor is able to identify the contributory factor, and at Level 4, the assessor considers that it is extremely clear what the risk factor is, even to an untrained person.

At the conclusion of the assessment for each pen, the assessor was asked to provide three suggestions to reduce the level of risk for biting in that pen.

## 2.2 Assessor training

Assessors were trained in using the risk assessment tool via a day long training course co-ordinated by Animal Health Ireland (AHI). This was carried out as part of the Targeted Advisory Service on Animal Health, part of the Rural Development Plan 2014–2020, funded by the Department of Agriculture, Food and the Marine (the competent authority for on-farm animal welfare in Ireland) in conjunction with the EU. The training programme also met the criteria for providing continuous professional development points granted by Veterinary Council of Ireland. Two of the authors of this paper were involved in delivering the training (CCG and KOD). The course consisted of theoretical (lecture based) and practical (pen side) training, and incorporated elements of interactive and cooperative learning [16].

The theoretical section started with introducing the requirements of Council Directive 2008/120 EC, information regarding risk assessment and pig behaviour, and finally an explanation of the structure of the adapted risk assessment tool. The practical aspect of the training

**Table 2. Levels of risk that the assessors were asked to assign to each risk category as described in Table 1.**

| Level of risk | |
| --- | --- |
| 0 | Risk not observed |
| 1 | Risk Category Statement is correct |
| 2 | I was not able to identify risks associated with this Risk Category |
| 3 | I have identified that risk exists for this Risk Category |
| 4 | There are clearly risks associated with this Risk Category |

took place at the Teagasc Moorepark pig facility, whereby small groups (2–4 veterinarians) observed four pens: two at weaner stage and two at finisher stage. At the first pen in each stage participants were shown how to, and practiced, observing animals for the six physical welfare indicators in the risk assessment tool, as well as the performance of all occurrence behaviour observations. At the second pen they carried out the assessment themselves. Once observations were completed, the group returned to the meeting room to discuss their results.

## 2.3 Application of the tool on commercial farms

Once assessors had completed the training, they were authorised by AHI to use the risk assessment tool on commercial pig farms. Assessments were carried out upon the request of the producer, and were conducted by their own private veterinary practitioner. Data collection by assessors for this study started in November 2019. Once a risk assessment was complete, the assessor uploaded the data into AHI's online tail biting risk assessment database, and informed staff at AHI whether the producer was willing to take part in this study. If so, and immediately after the data input, the research team based in Teagasc Moorepark received an email from AHI informing them of the producer and farm name, their location, and the name of the assessor. The producer was then contacted, and if willing to continue with participation, a date was arranged for the main author of this paper (RDA) to inspect a batch of their pigs for tail lesions at the abattoir within a month of the farm assessment. This timeline was established to ensure that pigs assessed at the finisher stage using the developed tool could also undergo post-mortem tail lesion scoring at the abattoir.

## 2.4 Assessment of pigs' tails in the abattoir

A total of 27 pig producers across the island of Ireland granted permission for the examination of a designated batch of their pigs at the abattoir (Table 3). The number of farms investigated varied across the years, with n = 2 farms in 2020, n = 11 farms in 2021, and n = 14 farms in 2022. Upon obtaining consent from the producers, a batch of pigs sent to the abattoir from their respective farms was subjected to inspection. This process aimed to ensure that carcases assessed in the abattoir were representative of the pig population that was assessed using the assessment tool on farms. The average duration between the initial farm assessment and the evaluation of skin tail lesions at slaughter was determined to be 32 ±19.8 days (mean ± s.d),

**Table 3. The geographical location by county of the Republic of Ireland of the 27 farms assessed using the risk assessment protocol, and their respective abattoir.**

| Farms County | Total assessed | Goegraphical location of abattoir by County | | | |
|---|---|---|---|---|---|
| | | Cork | Offaly | Tipperary | Waterford |
| Carlow | 4 | | | | ✓ ✓ ✓ ✓ |
| Cavan | 1 | | ✓ | | |
| Cork | 4 | ✓ | | ✓ ✓ | ✓ |
| Kerry | 1 | | ✓ | | |
| Kilkenny | 2 | | | ✓ ✓ | |
| Laois | 2 | | | | ✓ |
| Limerick | 1 | | | ✓ | |
| Meath | 3 | | ✓ ✓ ✓ | | |
| Tipperary | 1 | | | ✓ | |
| Waterford | 2 | | | | ✓ ✓ |
| Wexford | 6 | | | ✓ ✓ | ✓ ✓ ✓ ✓ |

and the study population (n = 7197 carcases) comprised pigs slaughtered at four abattoirs in the Republic of Ireland.

During each visit to the abattoir, a trained observer collected the following data from all pig carcases in the batch: sex, skin damage to the tail, presence/absence of bruises, and whether there was severe tail loss with healing. Data collection took place after scalding/dehairing operations using visual examination only (tails were not palpated or manipulated by hand). Skin damage was classified into five levels according to severity: 0 = no evidence of damage; 1 = minor skin damage to the tail tip without teeth marks; 2 = evidence of teeth marks, with breakage to the skin and redness; 3 = breakage of the skin with redness and swelling; 4 = fresh partial or complete tail loss, an open wound on the tail accompanied with pus or necrotic tissue. Additionally, severe tail loss with healing, defined as a completely missing tail, without open lesions, was recorded if present, representing a history of tail biting on the farm [17] (Fig 1). The presence or absence of bruising was recorded using a two-level scoring method: 0 = absence of bruises; 1 = presence of bruises (Fig 2) [17].

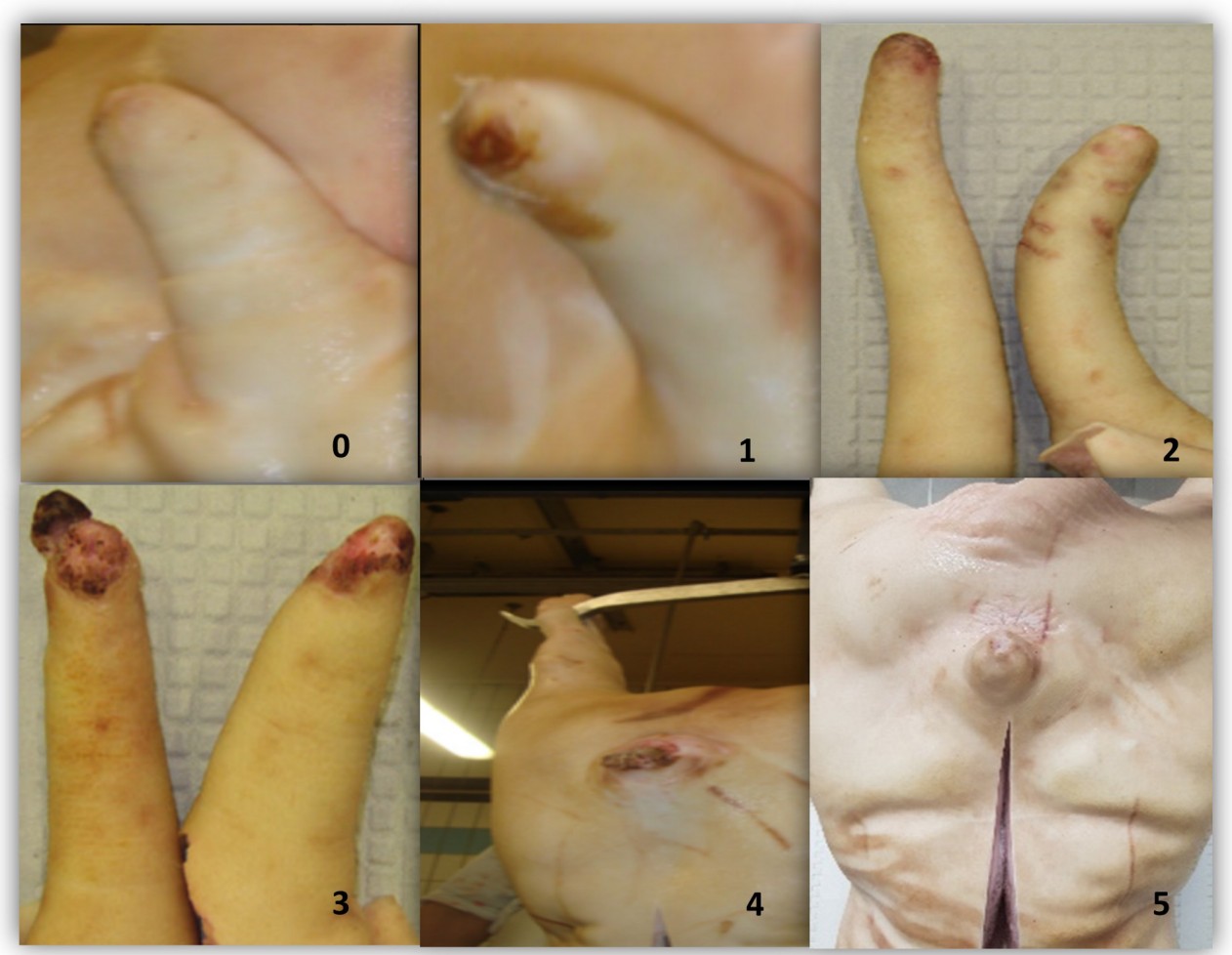

**Fig 1. Tail skin damage scoring system.** 0 = No evidence of tail biting; 1 = minor skin damage to the tail tip without teeth marks (mild lesion); 2 = evidence of teeth marks, with breakage to the skin and redness (mild lesion); 3 = breakage of the skin with redness and swelling (moderate lesion); 4 = fresh partial or complete tail loss, an open wound on the tail accompanied with pus or necrotic tissue (severe lesion); 5 = severe tail loss with healing.

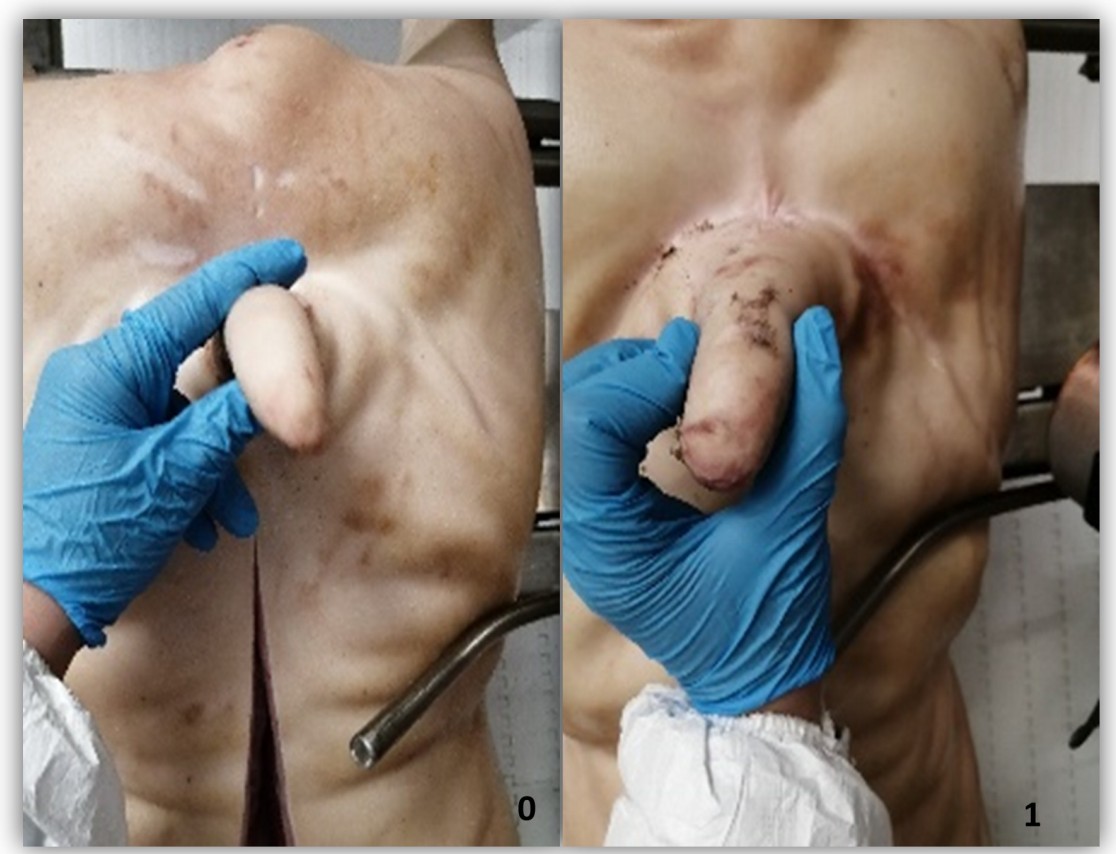

**Fig 2. Bruises detection scoring system.** 0 = absence of bruises; 1 = presence of bruises.

## 3. Descriptive statistics and data management

Data regarding management practices, enrichment provision and risk categories were summarized across all assessed pens. Due to the similarity between definitions, and low number of pens assigned to risk categories 0 and 4, assessor opinions on risk category were condensed into 3 Levels: Risks not observed (previously risk Level 0); Low risk of tail biting (previously risk Levels 1 and 2); High risk of tail biting (previously risk Level 3 and 4) (Table 1). The data was condensed in this fashion upon the advice of the third author of the manuscript, who has responsibility for liaising with assessors and co-ordinating training. Assessors were consulted, and they informed the authors that they considered both Level 1 and 2 to imply that there was minimal risk for tail biting in the pen. Levels 3 and 4 were being applied interchangeably whenever a risk factor related to that risk category could be clearly identified (e.g. a low enrichment allowance, is a clear risk for 'Enrichment'; single space feeders are a clear risk factor related to the category 'Feeding processes').

Whether or not the floor area was legally compliant was calculated by considering the area of the pen, the number of pigs present, and the final weight achieved in the pen. The results obtained were then compared to the current minimum space requirements in Ireland according to what is reported in the pig welfare requirements [18].

The proportion of pigs in each pen with each of the physical welfare indicators was calculated by dividing the number of affected pigs by the total number in the pen. Another variable, 'Visible injuries' was created by summing the number of tail, ear, flank and aggression lesions in the pen, and dividing by the number of pigs per pen to provide a pen level estimate relative to the number of pigs. The number of times each behaviour on the ethogram was performed was also divided by the number of pigs in the pen, and three new variables were also created: damaging (tail, ear and other biting), exploratory (exploration of 'fixtures and fittings' and of enrichment) behaviours, and the ratio between performance of exploration of enrichment and 'fixtures and fittings'.

The prevalence of tail lesion scores at the abattoir were calculated for each farm.

The management practices and welfare indicators recorded in the assessment that were considered to be associated with each risk category are described in Table 4. The decision as to which practices and indicators had potential to be related to each risk category was based upon the experience of the authors in the area of tail biting research. Only those that were associated with a risk factor were included in descriptive or statistical analysis.

## 3.1 Statistical analysis

Statistical analyses were conducted using SAS v9.3 (SAS Inst. Inc., Cary, NC). Each farm was considered an experimental unit. Proc Univariate was used initially to evaluate data distribution for all measures. Separate analyses, detailed below, were conducted using the data collected by the assessors to investigate 1) the relationship between management factors and welfare indicators, 2) the relationship between management factors and the level of risk assigned, 3) the relationship between welfare indicators and the risk of tail biting, and 4) relationship between level of risk assigned on farm and the tail-lesion scores in the slaughterhouse.

**3.1.1 Relationship between management factors and welfare indicators.** Due to the high number of pens where welfare indicators were not observed, analysis of individual

**Table 4. Management practices and welfare indicators considered to be associated with each risk category for tail biting defined in the assessment protocol.** The management practices and welfare indicators that are considered relevant for interpretation for each risk category are indicated by a tick mark directly across from them, under the relevant heading. Relevance was determined on the basis of the expertise of the project team. In italics management practices with the same response across all farms and that were, therefore, not analysed.

| | Risk Categories | | | | | |
| Management practices | Enrichment | Thermal comfort | Health | Competition | Pen design | Feeding processes |
|---|---|---|---|---|---|---|
| Enrichment | ✓ | | | ✓ | ✓ | |
| *Vaccination programme* | | | ✓ | | | |
| Space allowance | | | ✓ | ✓ | | |
| Feeder allowance | | | | ✓ | | ✓ |
| Drinker allowance | | | | ✓ | ✓ | |
| *Proportion slatted flooring* | | | | | ✓ | |
| **Welfare indicators** | | | | | | |
| *Tucked tails* | ✓ | | | ✓ | | |
| Injured tails | ✓ | | ✓ | ✓ | | ✓ |
| Injured/imperfect ears | ✓ | | ✓ | | | |
| Flank lesions | | | ✓ | ✓ | | |
| Aggression lesions | | | ✓ | ✓ | | ✓ |
| Dirty flanks/ haunches | | ✓ | | | ✓ | |
| Damaging behaviour | ✓ | | | | | ✓ |
| Aggressive behaviour | | | | ✓ | | ✓ |
| Exploratory behaviour | ✓ | | | | | |

indicators was not always possible. Separate linear mixed models (PROC MIXED) were used to evaluate the association between indicators and management practices. All of the models had the same set of predictors, with only the outcome variable changing between models. The predictor variables were: the number of enrichment items in the pen (0, 1, 2, and $\geq$ 3) and production stage (weaner stage 1, weaner stage 2 and finisher stage) were considered fixed effects, and stocking density (m2/pig) and the number of pigs per drinker considered continuous effects. Pen was considered a repeated effect within farm.

The dependent variables were: proportion of visible injuries, proportion of damaging behaviour, proportion of enrichment directed behaviour, proportion of 'fixture and fitting' directed behaviour, proportion of total exploratory behaviours, and the ratio between enrichment and 'fixture and fitting' directed behaviour. Because of the uniformity in the remaining management factors (Fig 3) it was not possible to include them in analysis.

**3.1.2 Relationship between management factors and level of risk assigned.** To evaluate the association between management factors and whether the assessor considered there was a risk for tail biting for that pen, as well as the level of risk associated with each relevant risk category, Spearman Correlations (PROC CORR) were performed when possible, and otherwise descriptive statistics reported.

**3.1.3 Relationship between welfare indicators and the risk of tail biting.** Physical welfare indicators were compared across pens that were considered to be at risk, or not at risk, using the non-parametric Wilcoxon Rank-Sum test (Proc Npar1way). All behavioural welfare indicators other than interaction with 'fixtures and fittings', and exploratory behaviour, were also compared in this way, as these two behaviours were observed in 99% and 100% of pens respectively. Analysis was carried out using a linear mixed model (PROC MIXED). Whether tail biting risk was considered to be present or not was considered a fixed effect, and pen was included as a repeated measure within farm (the experimental unit). A compound symmetry covariance structure was used.

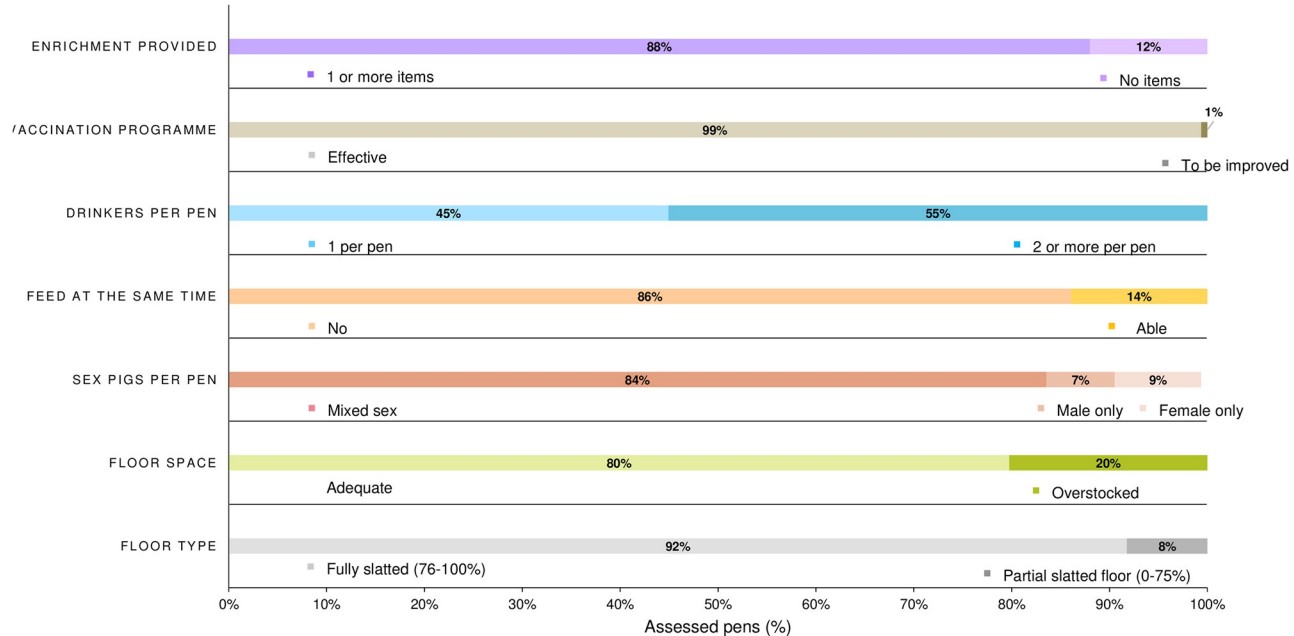

**Fig 3. Summary of management practices observed in all pens (n = 158) inspected:** The number of types of enrichment items provided in the pen (1 or more, no items); pigs in the pen have an effective vaccination programme (effective; to be improved); number of drinkers per pen (1; 2 or more); pigs can feed at the same time (no; yes); sex of the pigs per pen (mixed sex; males only; females only); floor space (adequate; overstocked); floor type (fully slatted (76–100%; partially slatted divided into 0–75%).

**3.1.4 Association between tail lesion scores post-mortem and level of risk assigned on farm.** Spearman rank correlation was used to investigate the relationship between the proportion of tail lesions in each slaughterhouse batch at each level of the tail lesion scoring system, and the proportion of tail lesions detected on farm. The association between the risks of tail biting and the proportion of tail lesion scores detected at post-mortem examination was then estimated using a linear mixed model (PROC MIXED). The level of tail biting risk was considered a fixed effect.

Results were considered statistically significant when α level was ≤ 0.05. For all predicted variables, residuals were checked for normality. Degrees of freedom were estimated using Kenwood-Rogers adjustment. Values are presented as LS means ± SE, unless stated otherwise.

# 4. Results

## 4.1 Farm description

Across the 27 study farms a total of 158 pens were included in the final cohort. These consisted of 85 finisher stage pens (37.4± 58.6 pigs/pen, mean ± s.d), 29 weaner stage 1 pens (60.1 ± 42.2 pigs/pen), and 44 weaner stage 2 pens (32.9 ± 13.7 pigs/pen). Although 6 pens were evaluated per farm, on some farms assessors had mistakenly inspected hospital or breeding animal pens, which were removed from the dataset. The total number of pigs included in the on farm inspections was 6421. The total number of pig carcasses inspected at the abattoirs was 7197 (158.9 ± 177.5 pigs/batch).

## 4.2 Management practices

Overall, 12% of pens had no enrichment provided, but the vaccination programme was considered appropriate in 99% (Fig 3). In 45% of pens there was only one drinker, and all pigs could feed simultaneously in only 14%. Pigs were separated by sex in 16% of pens. 20% of pens did not provide the minimum space allowance according to the legislation. Across all pens, 92% had a floor that was considered fully slatted (slatted area between 76–100%). The percentage of pens that contained pigs that were all tail docked was 92%.

## 4.3 Enrichment provision

Only 15% of the pens had a combination of different categories of enrichment items (Table 5). Among those, 11% had a combination of marginal and suboptimal, < 1% had marginal and

**Table 5. Summary of the number and classifications of enrichment materials observed in the 158 assessed pens.** The classification of enrichment materials was carried out according to the EU Commission's working document [14].

| Enrichment | % Pen | Marginal[1] | Suboptimal[2] | Marginal + Suboptimal | Marginal + Optimal[3] | Suboptimal + Optimal |
|---|---|---|---|---|---|---|
| None | 12% | . | . | . | . | . |
| 1 Item | 36.7% | 56 | 2 | . | . | . |
| 2 Items | 39.9% | 42 | 11 | 6 | | 4 |
| 3 or more Items | 11.4% | 4 | 1 | 12 | 1 | . |
| Number and type of items not reported | 0.63% | . | . | . | . | . |

[1]Marginal = objects like hard plastic piping or chains

[2]Suboptimal = peanut shells, ground wood, ground maize corn cobs, natural ropes, compressed straw cylinders, pellets, hessian cloth, shredded paper, or natural soft rubber as bedding, or optimal materials presented in a rack.

[3]Optimal = straw, green fodder (hay, grass, silage, alfalfa, etc.), miscanthus pressed or chopped, and root vegetables when used as bedding.

**Table 6. The proportion of times each level of risk was assigned to each of the risk categories by the assessors (n = 158 pens).**

|  | Enrichment | Thermal comfort | Health | Competition | Pen design | Feeding processes |
|---|---|---|---|---|---|---|
| No Risk | 4% | 5% | 9% | 1% | 5% | 9% |
| Minor Risk | 9% | 84% | 76% | 84% | 87% | 79% |
| Major Risk | 87% | 11% | 15% | 15% | 8% | 12% |

optimal, and 3% had sub-optimal and optimal. In total, 37% of the assessed pens were only provided with one type of enrichment: suboptimal or marginal.

## 4.4 Risk categories

Out of the 158 pens, 91 (58%) were classified by the assessors as being at risk for tail biting, while 65 (41%) were classified as not at risk. For 2 pens (1%), the risk level was not indicated by the assessors.

The majority of the time, assessors assigned a level of 'minor risk' for tail biting for the categories thermal comfort, health, animal competition, pen design and feeding processes (Table 6). However, when considering enrichment provision, assessors most often assigned a level of 'major risk'.

Although 41% (n = 65) of pens assessed were considered to have no overall risk of tail biting by the assessors, in these pens, assessors often assigned either a minor, or both minor and major, level of risk associated with each of the specific risk categories (Table 7). Out of these pens, 80% were categorised as having a major risk for the risk category 'Enrichment'. The only risk category for which assessors did not assign any level of major risk in these pens, was for 'Feeding Processes'.

## 4.5 Tail skin damage

At slaughter, 69.6% of the pig tails presented no tail damage (score 0), 15.5% presented minor skin damage to the tail tip (score 1), 11% presented evidence of teeth marks (score 2), 3% presented moderate damage (score 3), and only 1% presented severe tail skin damage (score 4). At farm level, minor skin damage was always most prevalent, and ranged from $\geq$ 83% to 100% of carcasses. Moderate damage ranged from $\geq$1% to $\leq$ 14% of carcasses, and severe damage from $\leq$1% to $\leq$4% of carcasses.

## 4.6 Relationship between management factors and welfare indicators

There was no effect of any of the management factors on the proportion of visible injuries, or on performance of damaging behaviour.

Both the stage of production and the number of enrichment items in the pen affected enrichment directed behaviour ($F_{2,135} = 4.6$, P = 0.01; $F_{3,144} = 10.37$, P < 0.001, respectively), and general exploratory behaviour ($F_{2,136} = 2.95$, P = 0.05; $F_{3,143} = 3.01$, P < 0.05, respectively).

**Table 7. The proportion of times each level of risk was assigned to each of the risk categories by the assessors, in the sub-sample of pens where there was considered to be no overall level of risk (n = 65 pens).**

|  | Enrichment | Thermal comfort | Health | Competition | Pen design | Feeding processes |
|---|---|---|---|---|---|---|
| No risk | 11% | 12% | 23% | 3% | 12% | 22% |
| Minor risk | 9% | 82% | 66% | 80% | 75% | 78% |
| Major risk | 80% | 6% | 11% | 17% | 12% | 0% |

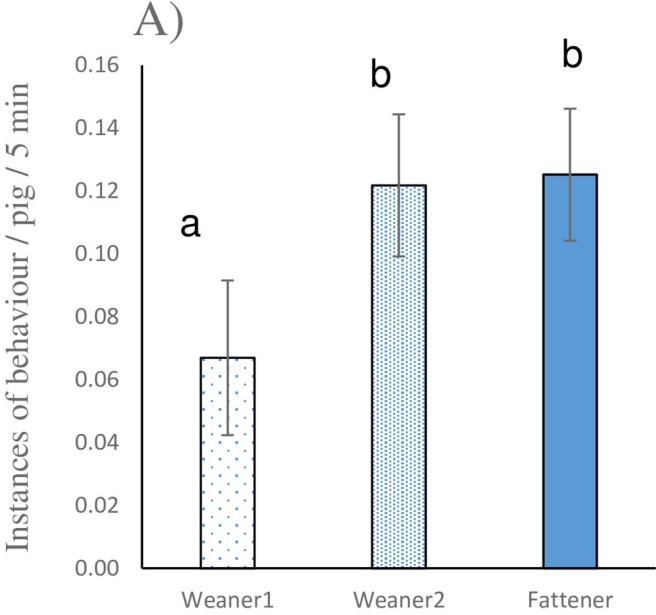

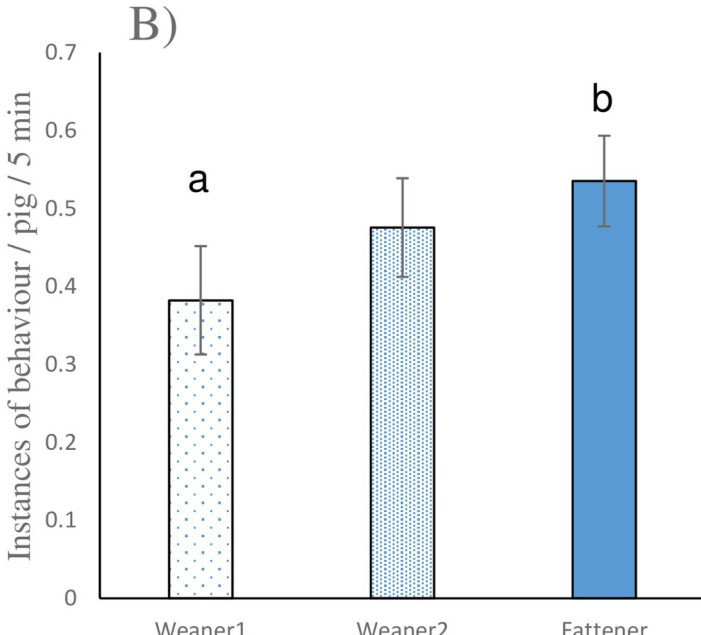

**Fig 4.** A) Enrichment directed behaviours, and B) general exploratory behaviours, performed per pig per 5 minute observation period, at the various stages of production. a, b indicates a significant difference. Data are presented as least squares mean ± s.e.

Both enrichment and exploratory behaviour increased in frequency with pig stage (Fig 4A and 4B). In general, enrichment directed behaviour increased as the number of items provided increased (Fig 5). There was no clear pattern when it came to the relationship between the number of enrichment items provided and exploratory behaviour, but the amount reported was numerically highest when one item was provided, and this was significantly higher than

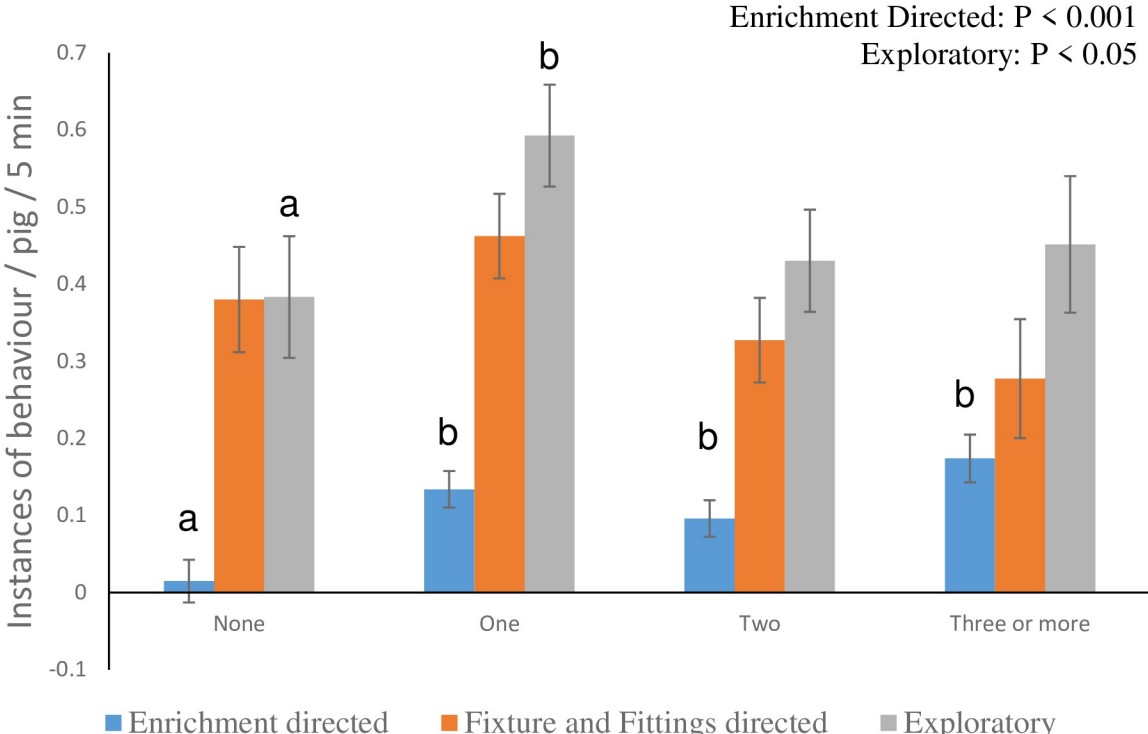

**Fig 5. Enrichment directed, 'fixture and fitting' directed, and overall exploratory behaviours performed per pig per 5 minute observation period, relative to the number of enrichment items provided in the pen.** a, b indicates a significant difference. Data are presented as least squares mean ± s.e.

when there were no items ($T_{141}$ = 2.43; $P < 0.05$). Although not statistically significant, there was a numerical indication that there were fewer interactions with 'fixtures and fittings' as the number of enrichment items increased (Fig 5). The only other management factor that had an impact on any welfare outcome was the number of drinkers per pig; as the proportion increased, so did exploratory behaviour (Regression coefficient: 0.825; $F_{1, 150}$ = 4.16; $P < 0.05$;).

## 4.7 Relationship between management factors and level of risk assigned to each risk category

The management factors that were investigated in association with each risk category are outlined in Table 4. A negative correlation was found between the number of enrichment items provided and the level of risk assigned to the risk category 'Enrichment' ($\rho$ = -0.35; $P < 0.01$). It was not possible to correlate any other management factors with the level of risk associated with the risk categories, as the data were often similar for each assessment, and as such descriptive data are provided below.

We did not consider any of the management factors that were assessed to be associated with a risk for Thermal comfort, and as such no data are reported. For the risk factor 'Health', the vaccination protocol was considered to be appropriate in 99% of pens, and as such this did not appear to contribute to the assessors considerations. A major level of risk was assigned to only 15% of assessed pens, even though space allowance did not meet legislative requirements in 20% of pens.

The risk category 'Competition' was assigned a level of 'Major risk' in only 15% of pens, even though 37% of the assessed pens had only one enrichment item, 20% had insufficient floor space, 86% of pens pigs could not feed at the same time, and 45% pens had only one drinker.

For the risk category 'Pen Design', a level of 'Major risk' was only assigned to 8% of pens, even though as described above, enrichment and drinker allowance was low, and 92% of pens had fully slatted floors.

The risk category 'Feeding processes, was considered a 'Major' risk factor in 12% of the pens assessed, although 86% of pens provided insufficient feeding space for pigs to feed at the same time.

## 4.8 Relationship between welfare indicators and risk of tail biting

**4.8.1 Physical welfare indicators.** None of the physical welfare indicators were individually observed in more than 45% of the assessed pens, but when the presence of any visible injuries was considered, 65% of pens had at least one pig affected (Table 8). The median and interquartile range of the proportion of pigs in pens where risk was and was not considered present were close to 0 in all cases. Despite this, pens that were not considered to be at risk of tail biting contained more pigs with flank lesions, and aggression lesions, than those that were not considered to be at risk (Table 8).

**4.8.2 Behavioural welfare indicators.** The only two behaviours that differed in frequency between pens considered at risk, or not at risk of tail biting were enrichment directed behaviour ($Z = 2.09$; $P < 0.05$) and aggressive behaviour ($Z = 2.12$; $P < 0.05$). In both cases, more of these behaviours were observed in pens that were not considered at risk of tail biting (Enrichment: 0.13 (0.04–0.21); Aggressive: 0 (0–0.03); median (interquartile range)) than in pens that were considered at risk of tail biting (Enrichment: 0.06 (0.01–0.15); Aggressive: 0 (0–0); median (interquartile range)).

**4.8.3 Association between the risk of tail biting and tail condition post-mortem.** The proportion of tail injuries/pig/pen at farm only correlated with the proportion of each batch in the abattoir that were categorised at level 4 (fresh partial or complete tail loss, an open wound on the tail accompanied with pus or necrotic tissue) according to the tail lesion scoring system ($\rho = 0.15$; $P = 0.05$).

In all cases assessors consistently reported that all 6 pens on each farm were either at risk, or not at risk, for tail biting. Thus each farm was classified as being either at risk or not. There was no difference between farms considered at risk, or not at risk of tail biting with regard to the prevalence of all tail skin damage considered (Fig 6).

**Table 8. The proportion of pigs in pens affected by physical welfare indicators in pens where private veterinary practitioners considered that there was either no risk or a risk of tail biting.** Data presented as median and interquartile ranges.

|  | % pens affected | No risk | At risk | Z | P- value |
|---|---|---|---|---|---|
| **Tucked tails** | 15% | 0 (0–0) | 0 (0–0) | 1.25 | 0.21 |
| **Injured tails** | 23% | 0 (0–0) | 0 (0–0) | 0.28 | 0.78 |
| **Injured/imperfect ears** | 39% | 0 (0–0.05) | 0 (0–0.06) | 0.98 | 0.33 |
| **Flank lesions** | 31% | 0 (0–0.02) | 0 (0–0) | 3.32 | <0.001 |
| **Aggression lesions** | 19% | 0 (0–0.05) | 0 (0–0) | 2.29 | 0.02 |
| **Dirty flanks/haunches** | 45% | 0 (0–0.1) | 0 (0–0.23) | 0.09 | 0.92 |
| **Visible injuries** | 65% | 0.07 (0–0.2) | 0.04 (0–0.17) | 0.78 | 0.43 |

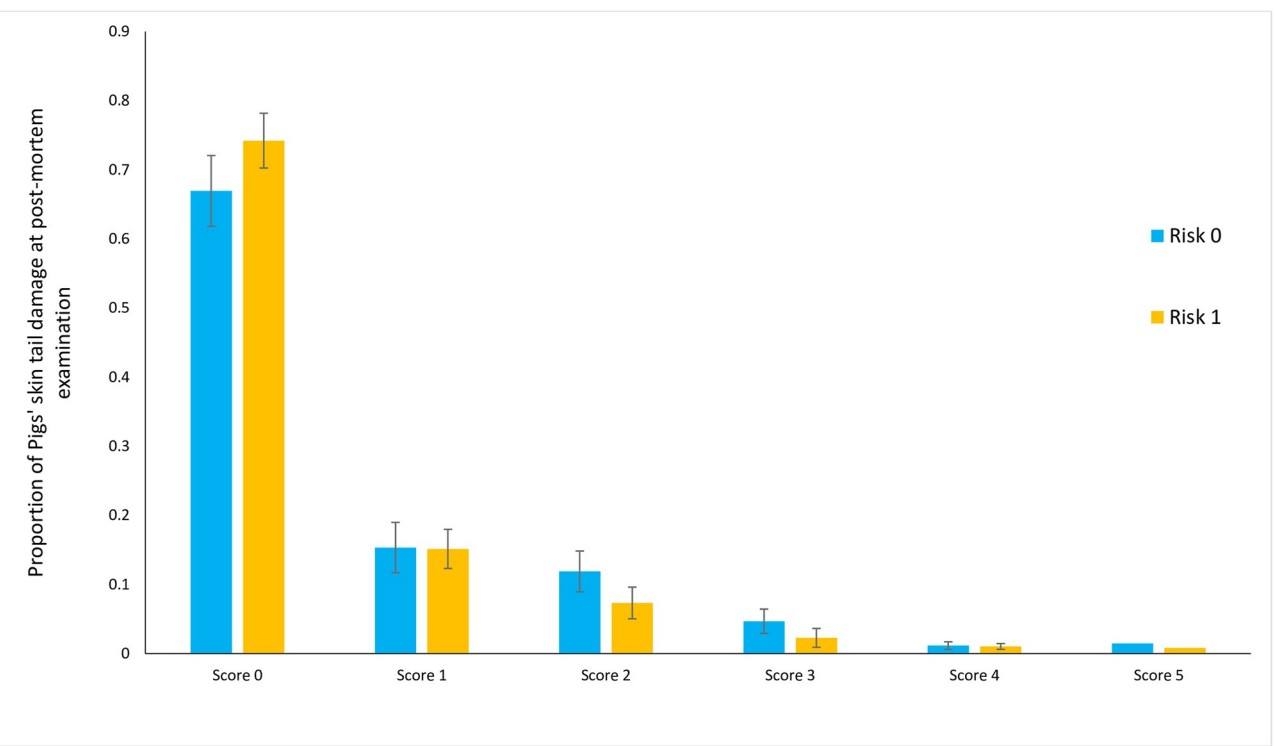

**Fig 6. Relationship between the proportion of tails with minor, moderate and severe tail skin damage detected at post-mortem examination, relative to whether farms were considered at risk, or not at risk of tail biting.** The data are presented as least squares means ± s.e.

## 5. Discussion

An assessment and control tool for tail biting is currently required for each EU Member State. To comply with this requirement in Ireland, a tail-biting risk assessment tool, adapted from a previously developed one and which was designed to be simple and quick to use, and suited to Irish commercial pig systems, was used to assess tail biting risk via a national scheme. It included consideration of management strategies, resource provisions and animal-based indicators, to provide guidance for the assessor in their perception of the risk of tail biting in a pen, and in aiding identification of specific risk factors at pen level. We used assessments collected as part of this scheme, along with detailed tail inspections at the slaughterhouse, to assess whether the tool is effective. One significant issue that the assessments identified is that tail docking is still the main strategy used to reduce the risk of tail biting in Irish pig production systems. The tool was also effective in identifying specific management strategies that should be, but are not currently, implemented according to legislation (e.g. provision of appropriate environmental enrichment). However, overall, we identified that the current application of the tool is ineffective in facilitating an association between management practices and animal based welfare indicators, and the level of risk for tail biting, not only on farm, but also relative to detailed tail lesion scoring at the abattoir.

As well as animal-based indicators, management and resource-based indicators are also important in risk assessment for tail biting [2]. This is because of its multifactorial aetiology, which incorporates risks associated with internal (age, and sex of the animal), and external (pen design, thermal conditions etc.) factors. Thus an effective risk assessment tool requires evaluation of management practices and resource provision to identify potential hazards and

allow recommendations to reduce risk. Management strategies and resource provision identified in this study, which are well recognised in the literature as being risks for tail biting, included widespread fully slatted flooring, low floor space allowance, lack of possibility to feed at the same time, a low number of drinkers per pen, and insufficient or appropriate enrichment provision [19]. Indeed the results showed areas of non-compliance with Council Directive 2008/120/EC, which regulates the minimum flooring space per pig and provides guidelines for enrichment provision; a high percentage of pens provided inadequate space per pig, and similarly, in most pens there was an absence of suitable environmental enrichment. It is well known that animals subjected to overcrowding can experience social stress [20], which may manifest as aggression, tail biting, and also impede productivity (EFSA 2022). The interaction between floor space, feeder space and pig production has been widely studied [21–23], with data suggesting that insufficient behavioural space in pens may negatively impact the ability of pigs to access the feeder and, consequently, impair their growth rate [19]. However, in reality at farm level, maintaining a low stocking rate within the legal limit can be complicated by genetic selection for increased output for sow per year. For example litter size increased from 10.8 in 2000 to 13.5 in 2017 in Ireland [24], and thus more pigs are being produced, but without necessarily a corresponding increase in facility size. In a previous study conducted in Ireland in 2019, it was suggested that to prevent overcrowding and mitigate the risk of tail biting, it may be necessary to reduce the number of breeding sows if housing facilities cannot be expanded [25].

The provision of appropriate environmental enrichment is important so that pigs have an outlet to perform highly motivated exploratory behaviour, reduce stress, and prevent the development of injurious behaviours such as tail and ear biting [26]. Commission recommendation 336 of 2016 provides guidance as to which materials to use, to ensure that it is effective and beneficial [15]. However the results show, where enrichment was provided, there was widespread use of long lasting and easy to maintain material, such as hard plastic toys, which are considered of marginal benefit to pigs. Such a management strategy, to favour long lasting enrichment items in order to reduce labour which would otherwise be required for replenishment, is likely to be counterproductive and financially costly [27]. It is also the case that on farms with fully slatted floors, farmers are hesitant to use straw or other fibrous enrichments that could block the manure system [15, 28–30].

Accordingly, the assessors recognised that lack of enrichment provision increases the risk of tail biting, and considered this a major risk in 87% of pens. The negative correlation between this risk category and the number of items provided, demonstrates that the assessors realised that when enrichment provided by farmers was insufficient, tail biting risk increased. At the same time, our results are in line with a previous study conducted in Ireland, which reported that using enrichment material alone is not enough to avoid the risk of this damaging behaviour [31], and other factors should be considered [19].

Unfortunately, the level of risk assigned to the other categories did not reflect the management strategies reported by the assessors. It is possible that the low number of factors that are included in the tool limits its usefulness in terms of assessing/attributing risk for tail biting relevant to the assessed farm. However, what also seems likely is that the assessors did not recognise potential hazards, as they were relatively inexperienced with the concept of risk assessment for tail biting. The subjective nature of the assessment tool, whereby assessors were required to use their judgement in assigning risk level, without specific guidance, is likely one of the reasons why the tool was found to be ineffective in aiding the assessors to identify specific risks.

In assessing animal welfare and the risk of tail biting, scoring of injuries and skin damage are robust indicators [26]. Skin lesions on the front part of the pigs body reflect high fighting frequency among individuals, whereas skin damage on the rear part (including tails) indicates

that individuals are recipients of injurious behaviour [32]. Body lesions were assessed from outside the pen, as is usual for veterinarians and stockpersons [33, 34], but the severity was not considered. Though this approach has been used in previous studies, it is challenging and presents limitations due to variation in group size and the possibility of pigs laying down and huddling [33]. Contrary to expectations, our results reveal that the prevalence of visible injuries remains unaffected by the production stage or the provision of enrichment materials. This contradicts certain studies suggesting that the manifestation of these welfare indicators varies with the age of the animals. Notably, the visibility of injuries may be contingent upon the size of the animals, with smaller piglets potentially inflicting less severe bites [35] or exerting reduced pressure on pen mates' bodies during mounting, thereby diminishing the likelihood of visible wounds. It is possible that due to the method of scoring used during the assessment, many injuries were simply not observed. We can also speculate that in younger animals, body lesions are more difficult to see due to their smaller bodies. It also must be noted that the way the data was collected did not permit calculation of the median number of lesions per animal, or what the distribution was per pig. The average number per pen, although useful, does not distinguish between pens which could contain few pigs with a large number of lesions, or pens where most pigs have a low number, which could provide additional insight into risk factors for biting.

A further finding concerning injuries is that the assessors did not necessarily associate these with the risk of tail biting behaviour; pens considered not at risk, had a higher proportion of pigs with flank and aggression lesions. A possible explanation is that the training provided was insufficient to enable assessors to carry out observations reliably for both body lesions and behaviour. In fact earlier studies proposed that for training to be effective, a minimum workshop duration of three days is recommended. This should involve practical training incorporating interactive learning methods, including the use of photos and videos [36]. We can also hypothesise that the conditions on farm (such as light level, overcrowding, etc,) also limited the assessor's ability to detect lesions. The production stage did not influence the observed performance of damaging behaviours, which is in contrast to other studies that have shown aggressive and harmful behaviour to decrease with age, as pigs tend to adapt to the group and their environment [23, 37, 38].

As expected, the use of enrichment influenced the performance of exploratory behaviour. In this study, exploratory behaviour was defined as the sum of the observations of pigs manipulating the enrichment material and pen fixtures and fittings. Pigs housed in pens with one or more enrichment item demonstrated a higher frequency of exploratory activities than those in pens without such items. While not statistically significant, our results mirror patterns documented in the literature, indicating that pigs in barren environments tend to manipulate pen fixtures more than those housed in enriched environments [39]. This behaviour is attributed to the fact that the exploration of pen fixtures and bare flooring is less gratifying than interactions with manipulated enrichment materials, even if not consistently categorised as 'chewable,' 'destructible,' 'rootable,' and 'deformable' materials conducive to enabling pigs' exploratory behaviours [40]. Moreover, these data suggest that the use of enrichment material could reduce competition for access to fixtures and fittings such as drinkers. This is in line with previous research that demonstrated that provision of a rack of optimal enrichment decreased drinker usage relative to pens without the rack [41].

We found an association between aggressive and enrichment-directed behaviour and pens assigned as no risk for tail-biting. The inexperience of the assessors with implementing the risk assessment tool was likely a potential barrier to assigning an appropriate level of risk. The requirement to conduct the assessment once a year per farm, and restrictions posed by biosecurity measures limiting visits to pig units to two farms per working week, may impede gaining proficiency with its implementation. One potential solution is to furnish assessors with

diverse online training materials, including recorded training courses. Additionally, regular testing of the assessors capability should be mandated between farm visits [36, 42].

The final method of assessing the robustness of the risk assessment tool was by comparing the presence of risk on-farm reported by the assessors, and the tail lesion scores obtained at post-mortem examination at the abattoir. The lack of association between these two parameters again suggests that in its current deployment, the application of the tool was not effective at identifying risk in general, even without considering the six specific risk categories. This was the case even though there was a positive correlation between the proportion of bitten tails observed on farm, and the proportion of tails that were highly damaged on the slaughter line. Thus although the assessors appeared to be identifying where tail lesions were prevalent on farm, this was not translating to the level of risk assigned. The condition of pig tails is considered an iceberg indicator for assessing animal welfare at slaughter [4, 43], and our finding is in line with other studies that have reported a link between observations reported on the farm and the condition of tails at the post mortem inspection [44].

There are a number of reasons that could have contributed to the lack of a link between the lesion prevalence recorded at the abattoir with the level of risk assigned by the assessors. Firstly, it is possible that in some cases pigs with severe tail lesions could have died on the farm, or have been euthanised prior to being transported to the slaughterhouse; this management practice would not have been identified by the tool, and it would mean that the proportion of severe tail lesions observed in the abattoir would be lower than that observed on the farm. An alternative hypothesis is that it could be easier to assess the presence of injuries to the tail during the post-mortem examination. Indeed, it is reported that scoring carcases at the abattoir can help to highlight prevalence of injuries, especially when these are widely recognisable by the presence of open wounds or necrosis [4]. Conditions on farm, namely overstocking of animals and dimly lit areas, may have impeded the observations. Finally, it is possible that the lack of experience of the assessors in identifying risk factors for tail biting could have reduced their ability to appropriately assign risk depending on management practices and animal based indicators.

There were a number of limitations associated with the study design. The inclusion of a relatively small number of farms in the study raises concerns about the generalizability of the findings to the broader national assessment programme. Moreover, results may differ if farms with pigs that were not tail docked were included. Future work could compare the efficacy of tail biting risk assessment protocols on populations of both docked, and undocked pigs. The potential lack of diversity in farm characteristics may restrict the applicability of the results to a more comprehensive context. At times, there was a significant temporal gap between the farm assessment and the subsequent visit to the slaughterhouse. This temporal lag introduces variability, as on-farm conditions may change during the intervening period. The resultant inconsistency in the relationship between factory and farm data across the study duration may thus impact the robustness of the conclusions drawn. As conditions can rapidly change in the farming environment this could influence the reliability of the observed relationships between factory data and on-farm variable across the study. Finally, the study's duration coincided with the COVID-19 pandemic, spanning approximately two years. This temporal context introduces the possibility that the assessors' skill levels may have evolved over the course of the study. Improved proficiency over time may affect the consistency and accuracy of data collection, potentially introducing a confounding variable that should be considered in the interpretation of results.

## 6. Conclusion

This study showed that the system that is currently in place to identify risk for tail biting in Irish pig farms, performs poorly. The adapted tool was developed following the guidelines in

the Commission Recommendation (EU) 2016/336) [15]. However, although the data, including the management section, helped identify the major issues present in the farming system, assessors were unable to link on-farm resource or management-based hazards to the risk of tail biting using the adapted tool. Limitations associated with the implementation of the tool include the relative inexperience of the assessors, and the challenge presented by recording animal indicators from outside of pens. Nevertheless, this study has identified areas for further study and, in particular the need to enhance the training of assessors, as they have an important advisory role for pig producers regarding animal health and welfare. Finally, the study also showed areas of non-compliance with EU legislation where further improvement is needed, namely the widespread use of tail docking, some overstocking, and suboptimal use of environmental enrichment.

In conclusion, our study underscores the critical importance of adopting a multifactorial approach to identify key management and resource risks that contribute to an elevated risk of welfare issues in pigs. Whilst tail biting is a complex, multifactorial behaviour, there is robust evidence from the international research community to demonstrate key hazards such as high stocking density, inadequate feeding space and sub optimal provision of environmental enrichment (e.g EFSA) [2, 26]. A number of studies [25, 34], including this one have identified hazardous management strategies in the Irish pig production system, which are in non-compliance with European regulations and in this regard may be considered as an 'accepted norm' within the industry. Such norms necessitate cultural change within the industry, working in partnership with other key stakeholders. In terms of risk assessment, adopting a co-design or similar collaborative approach in developing practice-ready tools is required. Such initiatives can foster a deeper understanding and enhance compliance with legislation at the national and international level [15, 45–47].

Furthermore, the study emphasizes the need for a more comprehensive training for assessors, to ensure a standardised approach to assessing risk. For example, the provision of webinars, and accompanied visits during a defined period. Despite the tool's inability to provide a reliable risk assessment for tail-biting, our research reveals that resource and management-based measures, coupled with data collected in abattoirs, can still yield robust information. This data effectively delineates the current risks of tail-biting on commercial farms in Ireland. Moving forward, these insights can serve as a compass to steer the refinement and development of more effective tools and strategies aimed at enhancing pig welfare in the industry.

Moreover, our research brings to light substantial disparities in the perception of optimal preventive measures and intervention strategies for tail-biting across farms. These variations are influenced not only by individual experiences within countries but also across different farming systems. Recognising these differences, our research team has developed a second risk assessment tool for Irish pig farms in partnership with stakeholders, which is currently undergoing validation. This tool aims to provide a more comprehensive evaluation, identifying both strengths and weaknesses in pig farms and will generate a detailed report for farmers. The ultimate goal is to facilitate targeted preventive measures, thereby mitigating the risk of tail-biting and promoting overall welfare in pig farming practices.

## Supporting information

**S1 File. Assessment and management of risk factors in tail-biting in pig production.**
(PDF)

## Acknowledgments

The authors would like to extend their thanks to managers and personnel at each of the abattoirs involved, all farmers given their permission to follow their batches in the abattoir and the Private veterinaries practitioners who had assessed the farms.

## Author Contributions

**Conceptualization:** Roberta Maria D'Alessio, Carla Correia-Gomes, Keelin O'Driscoll.

**Data curation:** Roberta Maria D'Alessio, Keelin O'Driscoll.

**Formal analysis:** Roberta Maria D'Alessio, Keelin O'Driscoll.

**Investigation:** Roberta Maria D'Alessio, Keelin O'Driscoll.

**Methodology:** Roberta Maria D'Alessio, Carla Correia-Gomes, Keelin O'Driscoll.

**Resources:** Carla Correia-Gomes.

**Supervision:** Alison Hanlon, Keelin O'Driscoll.

**Writing – original draft:** Roberta Maria D'Alessio.

**Writing – review & editing:** Roberta Maria D'Alessio, Conor G. Mc Aloon, Carla Correia-Gomes, Alison Hanlon, Keelin O'Driscoll.

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
