## [Decision Letter · Decision Letter 0]

18 Dec 2023

PONE-D-23-38802Evaluation of a tool to identify risks for tail biting in pigsPLOS ONE

Dear Dr. D'Alessio,

Thank you for submitting your manuscript to PLOS ONE. After careful consideration, we feel that it has merit but does not fully meet PLOS ONE’s publication criteria as it currently stands. Therefore, we invite you to submit a revised version of the manuscript that addresses the points raised during the review process.

**ACADEMIC EDITOR: **Please write a covering letter answering point-by-point the questions posed by the reviewers and mention the page number where the changes are implemented. Additionally, in the manuscript highlight those areas that have undergone changes. 

We look forward to receiving your revised manuscript.

Kind regards,

Faham Khamesipour, Ph.D.

Academic Editor

PLOS ONE

Journal Requirements:

2. You indicated that ethical approval was not necessary for your study. We understand that the framework for ethical oversight requirements for studies of this type may differ depending on the setting and we would appreciate some further clarification regarding your research. Could you please provide further details on why your study is exempt from the need for approval and confirmation from your institutional review board or research ethics committee (e.g., in the form of a letter or email correspondence) that ethics review was not necessary for this study? Please include a copy of the correspondence as an ""Other"" file.

5. In the online submission form, you indicated that "Data cannot be shared publicly because of GDPR considerations regarding farm audits. Data from the inspections at the abattoir are available upon request (contact via email Keelin.ODriscoll@teagasc.ie) for researchers who meet the criteria for access to confidential data."

Reviewers' comments:

Reviewer's Responses to Questions

**Comments to the Author**

1. Is the manuscript technically sound, and do the data support the conclusions?

Reviewer #1: No

Reviewer #2: Yes

Reviewer #3: Yes

Reviewer #4: Yes

2. Has the statistical analysis been performed appropriately and rigorously? 

Reviewer #1: No

Reviewer #2: Yes

Reviewer #3: Yes

Reviewer #4: Yes

3. Have the authors made all data underlying the findings in their manuscript fully available?

Reviewer #1: No

Reviewer #2: No

Reviewer #3: No

Reviewer #4: Yes

4. Is the manuscript presented in an intelligible fashion and written in standard English?

Reviewer #1: Yes

Reviewer #2: Yes

Reviewer #3: Yes

Reviewer #4: Yes

5. Review Comments to the Author

Reviewer #1: The submitted manuscript describing the evaluation of a tool to identify risks for tail biting in pigs fails to provide data to support the conclusions. The statistical analysis is not well described nor appropriate such that the results cannot be interpreted with any confidence. Therefore, any conclusions are not supported. There appears to be many instances of missing data.

Lines 282-288: the provided descriptive statistics are difficult to understand. Only the second paragraph refers to management practices, so move the heading (4.2 Management practices) and insert an additional heading 4.1 Descriptive statistics.

Example: does Line 283 indicate that the finisher stage sample population was 85 pens with 37.4±58.6 pigs/pen?

Example: what do the Line 287 numbers after 6421 and Line 288 numbers after 7197 refer to?

Lines 289-290: delete "A summary of the management practices observed over all pens can be seen in Figure 3."

Line 290: insert "(Figure 3)" after "99%"

Line 295: replace "that were all" with "tails"

Lines 298-299: delete "A breakdown of the category type and amount of enrichment observed in the pens is provided in Table 4."

Line 300: insert "(Table 4)" after "items"

Line 311: delete "Table 5 shows the level of risk assigned by the assessors with relation to each risk category"

Line 312: replace "minimum" with "minor"

Line 314: insert "(Table 5)" after "system"

Lines 315-316: delete "Table 6 reported the level of risk assigned for those pens where assessors reported no risk for tail biting."

Line 317: insert "(Table 6)" after "factors"

Table 5: please explain the percentages. Based on the title, values between 0 and 2 would be expected.

Line 326: change heading to "4.4 Skin damage"

Lines 327-330: Is the data presented somewhere in the manuscript?

Lines 334-337: Indicate the variation in the values that the statistics report. Reduce the 2 significant figures used in Line 335 to only 1 significant figures as in Line 337 for consistency.

Lines 338-339: If the data presented somewhere, please indicate. If not, please provide the values.

Line 350: Figure 5 only shows pig production stage, so remove "the amount of enrichment provision"

Line 353: more what?

Line 365: the data is not in Table 2

Line 375: indicate the association value

Lines 377-379: are Feeding System and feeding space confounded?

Lines 384-385: delete "Table 7 shows the number of lesions found by the assessors for each body condition listed in the risk

protocol, divided according to whether the risk of tail biting was detected."

Line 388: insert "(Table 7)" after "not"

Line 387: how was "no difference" determined?

Lines 397-398: Is the data presented somewhere in the manuscript?

Lines 417-546: were not carefully reviewed due to limitations in the results discussed previously

Figures 4-7. The standard error values seem very large and should be checked. All the values in the figures should be labeled to indicate statistical differences. Providing a single letter does not indicate which values are different from each other.

Reviewer #2: In their study, D’Alessio et al. Investigate the utility of a tail biting risk assessment tools for veterinarians. This tool includes a classification of several management and health xxx in combination with observations of behaviours to assess the risk of tail biting on farm. The risk assessment of trained veterinarians was compared to actual lesions recorded at the slaughterhouse and found to be not reliable. The authors provide a range of reasons why the tool failed in their article. Although the results provide a valuable overview of the prevalence of tail biting and reflect management practices in Ireland, they appear to be of limited interest to other countries with different assessment tools, particularly as the authors do not suggest any major improvements to the tool or training.

In general, the tool is insufficiently described. Without prior extended knowledge of the WebHat protocol, the protocol is difficult to understand. The authors should be clearer in explaining how the risk categories are assigned from the parameters evaluated. For each level of the parameter, it should be made clear how this impacts the assessment of risk, for example if there is only marginal enrichment how does this translate in assessment of risk value? Does marginal enrichment mean a risk of 4 in the category? What are the expectations regarding, for instance, vaccination status or mixing of sexes, and tail biting?

Throughout the results it is not enough to indicate (P<0.05) we need more information on the model output with information on the type of model, estimate, degrees of freedom, etc. This can be in the results section but also in a table. Same for the correlation outputs, just indicating rho values is not enough.

Material and methods

Line 119: The assigned values should be explained in the text and not only in the table. Please also give more details as to how the assessment was conducted for each category.

Table 1: “Risk 0: Risk not observed, Risk 1: Risk Category Statement is correct, Risk 2: I was not able to identify risks associated with this Risk Category, Risk 3: I have identified that risk exists for this Risk Category Risk 4: There are clearly risks associated with this Risk Category” this is very unclear and confusing and should be rephrased. I find this difficult to read and difficult to understand, I think adding in text explanations would help. For example, Risk 1 for the first category means “the risk category statement is correct” and the statement says “Environmental enrichment provision represents no risk for tail biting” so to me that means risk 1 there is no risk but risk 2 level says “I was not able to identify risks associated with this Risk Category” so risk 2 means there was no risk too? This is extremely confusing and need to be carefully explained. Putting values from 0 to 4 seems to hint that there is a hierarchical order but in that example I see no difference between risk 1 and 2.

Line 169: It is unclear to the reader what the sample size refers to, please clarify whether this is the number of animals per year or farm where you got permission for examination.

Line 212: consider using “grouped” instead of “collapsed”.

Line 213: there is one “and” too much in the parenthesis, consider replacing it with a comma

Line 220-221: Here, the associations between the risks and their indicators are presented; however in table 3 it is written “indicators hypothesised to be associated with each risk category” which is not specified in the text. I find this confusing as in the next line it says that only indicators associated with a risk factor were included in the analysis but it is unclear how these associations were decided and whether they are real or predicted.

Table 3: please provide information on how to interpret this table. It is not clear to me, for example, how dirty flanks are indicative of thermal comfort (or the lack of it).

Line 240 and 265: Please specify if the model was linear or generalized linear; in the latter case, which distribution was used.

Line 242: I assume these indicators were the response variable, but I feel this could be made clearer. To me it seems there was a separate model for each of these variables with management practices as fixed factors and pen as a random effect, is this accurate?

Line 245 and 246: factors instead of effects? I assume the number of pigs per drinker was also a fixed effect, but a continuous one.

Line 275: Tendency toward significance does not represent statistical significance and therefore should not be discussed as it can lead to over interpretation.

Line 277: The nature of the transformations needs to be specified for each variable they were necessary and for each model.

Results

Table 4: it is unclear what the categories mean, provide more detail in the text. For instance, is it the number or the type of enrichment objects that are classified as “suboptimal” etc.?

Line 308: What level of risk value does “considered to be a risk” refer to? It would be good to specify clearly if this is from risk 1, or 2. As I mentioned above, the risks levels are not well explained and make this difficult to understand. “assessors” instead of “assessor’s”.

Line 312: Please define in the methods what is the level of minimum risk. The risk category levels as explained in the methods are not clear enough to establish what is the minimum reported.

Table 5: Minor and Major risks need to be defined in the methods. From my example above I do not see how risk 1 is a minor risk from enrichment as risk 1 says the statement “Environmental enrichment provision represents no risk for tail biting” is correct. No risk for tail biting is not minor risk. Additionally, why are risk 3 and 4 not mentioned?

Table 6: does this table include the first row of Table 5? It is unclear what the table shows, and, in particular, what the difference is to Table 5.

Figures 4-6: difficult to interpret the letter, what are the comparisons? Why “P<0.05” in Fig. 7? There is no letter indicating statistically significant difference?

Line 368: provide p-value with coefficient

Discussion

The discussion is well written and I mostly agree with it. However, it is often mentioned that some observations and findings are contrary to the literature, because conditions prevented the assessors from measuring the variable correctly, or because insufficient training resulted in a lack of reporting of certain variables (such as injuries). In this case, however, this makes the evaluation rather unreliable, because if the assessors could not assess the different variables correctly, then it is difficult to draw any conclusions from this study.

It seems to me that the method was not standardized enough and allowed for too much interpretation by the assessors, which led to results that were ultimately uninformative.

Unfortunately, the discussion is confined to describing the situation and explaining the results, but do not suggest how to proceed or how to solve the problem, which would have made the manuscript of broader interest.

Reviewer #3: The paper is a very well written and understandable document that presents interesting data related to Evaluation of a tool to identify risks for tail biting in pigs, that fits the Scope of PLOS One.

In my perspective the study has some main limitations that should be improved by the authors:

- Please provide the assessment protocol (Supplementary material) and, more important, provide the typology of the answers to each evaluation made (e.g. qualitative, score, presence/absence....etc.)

- Line 93 “designed to take a maximum of 15 minutes to complete per pen by a trained assessor (S1).”- Please add how many pens need to be evaluated per farm (it is written after, but I think that it should be also add here)

- Line 98 - How the assessor obtained the final weight achieved in the pen?

- Line 100 101 : How the assessor considered the vaccination protocol to be appropriate? Plesae clarify

- Line 106: Evaluation of physical and behavioral welfare indicators. It is not clear how this evaluation was performed for each variable assessed (e.g. score for each injuries or presence/absence?). Please clarify.

- Line 116: “Estimate the level of risk of tail biting in relation to six risk categories.” This output it is extremely subjective. There was any score used to support the assessors? Please clarify

Line 120: “In each farm the assessor was asked to assess six pens; two at first stage weaner (from weaning day to approximately four weeks after weaning), two second stage weaner (from approximately four weeks after weaning to moving pigs to the finisher stage), and two finisher pens.” How the assessor selected the pens? The assessor evaluate the six pens in the same day? The risk estimated is a general risk per farm or per stage of production? The finisher pigs are the ones observed at abattoir? The pigs evaluated at the weaning stage are not the ones evaluated at abattoir? I’m asking this because the majority of tail biting outbreaks occurs at this stage and consequent severe tail loss with healing may be seen at abattoir. If these animals are not followed to the abattoir, some correlation analysis may failed. Please clarify these questions. Please explain the relationship between the batch of pigs sent to the abattoir with the assessed pigs at farm level

- Line 172 - Define representative sample

- Line 185 - Please define: severe tail loss with healing

- Figure 1: Photos from score 0, 1, 2 and 3 are not original. Please add the source

Line 486 - “ Our results indicate that the prevalence of pigs with the investigated injuries was low, and that visible injuries were not influenced by the production stage, or the number of enrichment materials provided per pen”. Please add some discussion regarding the fact that the majority were docked pigs which may have influenced the result you found. Please add the same information to the conclusions: Line 554 “ Limitations associated with the implementation of the tool include the relative inexperience of the assessors, and the challenge presented by recording animal indicators from outside of pens”

Line 520 - “The lack of association between the two again suggests that in its current deployment, the application of the tool was not effective at identifying risk in general, even without considering the six specific risk categories”. Please add some information/discussion about the possible bias if pigs with severe tail lesions at farm could have been euthanised or not included in the batch assessed at abattoir level.

Since Assessment of tail biting it is a topic of major importance in pork production chain, this submission may be suitable for publication, if authors improve the limitations aforementioned.

Reviewer #4: This study is very valuable because a significant reduction in the risk of tail biting can be achieved on many farms through the systematic evaluation and modification of management practices. The study aimed to validate a novel tail-biting risk assessment protocol by examining its association with tail lesion scores observed at slaughter. The authors indicated the inefficiency of the assessment tool in identifying actual risk, on farm. It underscored the need for enhanced training, and improvement at farm level in areas such as stocking density, enrichment provision and reducing tail docking, which are practical working solutions.

6. PLOS authors have the option to publish the peer review history of their article (what does this mean?). If published, this will include your full peer review and any attached files.

Reviewer #1: No

Reviewer #2: No

Reviewer #3: No

Reviewer #4: No

---

## [Author Response · Author response to Decision Letter 0]

16 Mar 2024

Review Comments to the Author

Reviewer #1: The submitted manuscript describing the evaluation of a tool to identify risks for tail biting in pigs fails to provide data to support the conclusions. The statistical analysis is not well described nor appropriate such that the results cannot be interpreted with any confidence. Therefore, any conclusions are not supported. There appears to be many instances of missing data.

AU: Thank you for your feedback and specific comments, and we hope that the manuscript is now easier to follow and review.

Lines 282-288: the provided descriptive statistics are difficult to understand. Only the second paragraph refers to management practices, so move the heading (4.2 Management practices) and insert an additional heading 4.1 Descriptive statistics.

AU: We have changed the headings to 4.1 Farm description, and 4.2 Management practices (Line 334).

Example: does Line 283 indicate that the finisher stage sample population was 85 pens with 37.4±58.6 pigs/pen?

AU: Yes, the line did indicate that over the total of 85 pens assessed in the finisher stage the average of pigs per assessed pens were 37.4±58.6 pigs per pen. This text has now been redrafted for clarity (Line 335).

Example: what do the Line 287 numbers after 6421 and Line 288 numbers after 7197 refer to?

AU: We have clarified the text in this paragraph (now L340 to 341).

Lines 289-290: delete "A summary of the management practices observed over all pens can be seen in Figure 3."

AU: The sentence was deleted accordingly.

Line 290: insert "(Figure 3)" after "99%"

AU: The sentence was updated accordingly. Now line 344.

Line 295: replace "that were all" with "tails"

AU: The sentence was changed to clarify that all pigs were tail docked (line 348).

Lines 298-299: delete "A breakdown of the category type and amount of enrichment observed in the pens is provided in Table 4."

AU: The sentence was deleted accordingly.

Line 300: insert "(Table 4)" after "items"

AU: The sentence was changed accordingly. The table becomes Table 5 (Line 352).

Line 311: delete "Table 5 shows the level of risk assigned by the assessors with relation to each risk category"

AU: The sentence was deleted accordingly.

Line 312: replace "minimum" with "minor"

AU: The sentence was changed accordingly (line 370).

Line 314: insert "(Table 5)" after "system"

AU: The sentence was changed accordingly (line 372). Now Table 6

Lines 315-316: delete "Table 6 reported the level of risk assigned for those pens where assessors reported no risk for tail biting."

AU: The sentence was deleted accordingly.

Line 317: insert "(Table 6)" after "factors"

AU: The sentence was changed accordingly (line 372).

Table 5: Please explain the percentages. Based on the title, values between 0 and 2 would be expected.

AU: Now Table 6. We have changed the wording of the title and hope that it is now clearer. “Table 6. The proportion of times each level of risk was assigned to each of the risk categories by the assessors (n = 158 pens)”

Line 326: change heading to "4.4 Skin damage"

AU: The heading was changed to “4.4. Tail skin damage” (Line 389). 

Lines 327-330: Is the data presented somewhere in the manuscript?

AU: The raw data are available as supplementary material. We have changed the wording of the paragraph as follow: “At slaughter, 69.6% of the pig tails presented no tail damage (score 0), 15.5% presented minor skin damage to the tail tip (score 1), 11% presented evidence of teeth marks (score 2), 3% presented moderate damage (score 3), and only 1% presented severe tail skin damage (score 4). At farm level, minor skin damage was always most prevalent, and ranged from ≥ 83% to 100% of carcases. Moderate damage ranged from ≥1% to ≤ 14% of carcases, and severe damage from ≤1% to ≤4% of carcases.” (lines 389-395)

Lines 334-337: Indicate the variation in the values that the statistics report. Reduce the 2 significant figures used in Line 335 to only 1 significant figures as in Line 337 for consistency.

Lines 338-339: If the data presented somewhere, please indicate. If not, please provide the values.

Line 350: Figure 5 only shows pig production stage, so remove "the amount of enrichment provision"

Line 353: more what?

AU: This section has been changed considerably in line with our revised statistics, and the results are now in graphical format (Figures 4 and 5). From line 396 to 422

Line 365: the data is not in Table 2

AU: Apologies for this typo, we meant Table 4. Now line 427.

Line 375: indicate the association value

AU: We have edited this paragraph so that the data presented reflects the management factors associated with the Risk Factor ‘Pen Layout’ (Lines 442-444).

Lines 377-379: are Feeding System and feeding space confounded?

AU: Now line 445. The two terms in fact refer to different things, and we hope that due to the significant re-writing we have carried out that this is now clear. ‘Feeding System’ refers to a ‘Risk category’, and ‘feeding space’ refers to the assessors’ evaluation as to whether or not the pigs can feed simultaneously. 

Lines 384-385: delete "Table 7 shows the number of lesions found by the assessors for each body condition listed in the risk protocol, divided according to whether the risk of tail biting was detected."

AU: The sentence was deleted accordingly. 

Line 388: insert "(Table 7)" after "not"

AU: This section has been re-worded considerably, and we hope that it is now more clear. From line 448 to line 456. Table 8 was then added to the section (line 458).

Line 387: how was "no difference" determined?

AU: This was determined by statistically comparing the proportion of animals that had physical impairments in each pen, in the pens that were deemed to have a risk of tail biting, and the pens that were deemed not to have a risk. This is described in the statistics section, L283-295. We have updated the results section to show the data and levels of significance.

Lines 397-398: Is the data presented somewhere in the manuscript?

AU: Apologies for the omission, these data are now included. We also noticed a slight error in the coding for the analysis, so some of the previously identified differences are no longer significant. Now lines 459-462

Lines 417-546: were not carefully reviewed due to limitations in the results discussed previously

AU: We hope that now that we have improved the materials and methods and statistics sections, that the reviewer will now feel confident in assessing the discussion. The discussion starts now at line 486

Figures 4-7. The standard error values seem very large and should be checked. All the values in the figures should be labeled to indicate statistical differences. Providing a single letter does not indicate which values are different from each other.

AU: We have updated the figures with the re-analysed data, and have improved their structure so that differences between means are more clearly indicated.

Reviewer #2: In their study, D’Alessio et al. Investigate the utility of a tail biting risk assessment tools for veterinarians. This tool includes a classification of several management and health xxx in combination with observations of behaviours to assess the risk of tail biting on farm. The risk assessment of trained veterinarians was compared to actual lesions recorded at the slaughterhouse and found to be not reliable. The authors provide a range of reasons why the tool failed in their article. Although the results provide a valuable overview of the prevalence of tail biting and reflect management practices in Ireland, they appear to be of limited interest to other countries with different assessment tools, particularly as the authors do not suggest any major improvements to the tool or training.

AU: Thank you for your comment. We have added new content to the discussion to address this issue. To support assessors and improve the training period, we suggested to prologue the training period, including the use online training material. An evaluation of the Private Veterinary Practitioners’ experience in implementing the tool would also support future adaptions of the tool, to improve its implementation (L600-602). We have also referred to a current study to adapt another more detailed Risk Assessment tool to assess hazards and protective factors on Irish pigs’ farms, and now we are in the process to validate it (L661-688). 

In general, the tool is insufficiently described. Without prior extended knowledge of the WebHat protocol, the protocol is difficult to understand. The authors should be clearer in explaining how the risk categories are assigned from the parameters evaluated. For each level of the parameter, it should be made clear how this impacts the assessment of risk, for example if there is only marginal enrichment how does this translate in assessment of risk value? Does marginal enrichment mean a risk of 4 in the category? What are the expectations regarding, for instance, vaccination status or mixing of sexes, and tail biting?

AU: The link to the WebHat is provided in the introduction at L69. To try and improve clarity further however we have added text to the materials and methods to explain better the definitions of the levels of risk, and how they should be applied (L134 – 141). It transpired that the assessors used the levels in a different way than instructed, and we have explained that from L235-241. This is why we grouped the risk levels in the analysis. It is discussed further in the discussion, and likely contributed to why the tool proved ineffective in identifying the specific risk factors for the assessed farms included in the study. (L549-552 and L560-602) 

Throughout the results it is not enough to indicate (P<0.05) we need more information on the model output with information on the type of model, estimate, degrees of freedom, etc. This can be in the results section but also in a table. Same for the correlation outputs, just indicating rho values is not enough.

AU: We have improved the information provided in the results section which now provides the information requested. (L333-480)

Material and methods

Line 119: The assigned values should be explained in the text and not only in the table. Please also give more details as to how the assessment was conducted for each category.

AU: The section was improved significantly, and we hope that the description of risk categories and levels of risk that could be applied are more clear (L134 to 152). 

Line 169: It is unclear to the reader what the sample size refers to, please clarify whether this is the number of animals per year or farm where you got permission for examination.

AU: The sentence was modified to clarify that the sample size refers to the number of farms assessed in each year (L192-194).

Line 212: consider using “grouped” instead of “collapsed”.

AU: This paragraph was re-written to improve clarity as to which welfare indicators were used in analysis (Line 238 - 244).

Line 213: there is one “and” too much in the parenthesis, consider replacing it with a comma

AU: See reply to previous comment. We have placed inverted commas around ‘fixtures and fittings’ throughout the manuscript to clarify that this was one item on the original ethogram.

Line 220-221: Here, the associations between the risks and their indicators are presented; however in table 3 it is written “indicators hypothesised to be associated with each risk category” which is not specified in the text. I find this confusing as in the next line it says that only indicators associated with a risk factor were included in the analysis but it is unclear how these associations were decided and whether they are real or predicted.

AU: The decision as to which indicators and management practices were selected was based upon the experience of the authors, which in itself is based upon the literature, and their own research into tail biting (L260-262). KO has managed a research programme in this area for 10 years, AH has previously supervised PhD and MSc students in the area of risk factors associated with tail lesions, and CG has extensive experience of consulting with Private Veterinary Practitioners and industry personnel in the area of risk factors for tail biting.

Table 3: please provide information on how to interpret this table. It is not clear to me, for example, how dirty flanks are indicative of thermal comfort (or the lack of it).

AU: Now Table 4. Dirty flanks were selected as a welfare indicator as pigs are known to huddle when too cold, and wallow when too warm, which can increase the risk of rubbing dirt on each others bodies. We have added some additional explanation to the legend.

Line 240 and 265: Please specify if the model was linear or generalized linear; in the latter case, which distribution was used.

AU: We have clarified that linear models were used. 

Line 242: I assume these indicators were the response variable, but I feel this could be made clearer. To me it seems there was a separate model for each of these variables with management practices as fixed factors and pen as a random effect, is this accurate?

AU: Yes, this is correct. We have re-phrased this paragraph and hope that you find it clearer (L283 – 295). 

Line 245 and 246: factors instead of effects? I assume the number of pigs per drinker was also a fixed effect, but a continuous one.

AU: See above 

Line 275: Tendency toward significance does not represent statistical significance and therefore should not be discussed as it can lead to over interpretation.

AU: We agree, and have deleted all reference to tendencies.

Line 277: The nature of the transformations needs to be specified for each variable they were necessary and for each model.

AU: No data were transformed, and as such we have deleted this sentence.

Table 4: it is unclear what the categories mean, provide more detail in the text. For instance, is it the number or the type of enrichment objects that are classified as “suboptimal” etc.?

AU: More clarification was added to the table in the form of footnotes. 

Line 308: What level of risk value does “considered to be a risk” refer to? It would be good to specify clearly if this is from risk 1, or 2. As I mentioned above, the risks levels are not well explained and make this difficult to understand. “assessors” instead of “assessor’s”.

AU: This refers to what is described in L144-152, whereby assessors were asked to consider whether they think there is an overall level of risk of tail biting in the pen. Grammar has been corrected.

Line 312: Please define in the methods what is the level of minimum risk. The risk category levels as explained in the methods are not clear enough to establish what is the minimum reported.

AU: We have added more description to the materials and methods regarding the levels of risk. It was up to the professional opinion of the assessor, without specific guidance, if not the statements reported in the tool, to assign the risk level. We agree that the definitions are not clear. However these were inserted by an external party who was also involved in the development of the tool, without piloting the definitions with either potential assessors or other personnel involved in the tool development.

Table 5: Minor and Major risks need to be defined in the methods. From my example above I do not see how risk 1 is a minor risk from enrichment as risk 1 says the statement “Environmental enrichment provision represents no risk for tail biting” is correct. No risk for tail biting is not minor risk. Additionally, why are risk 3 and 4 not mentioned?

AU: The data was condensed in this fashion upon the advice of the third author of the manuscript, who has responsibility for liaising with assessors and co-ordinating training. Assessors were consulted, and they informed the authors that they considered both level one and 2 to imply that there was minimal risk for tail 

---

## [Decision Letter · Decision Letter 1]

3 Apr 2024

PONE-D-23-38802R1Evaluation of a tool to identify risks for tail biting in pigsPLOS ONE

Dear Dr. D'Alessio,

Thank you for submitting your manuscript to PLOS ONE. After careful consideration, we feel that it has merit but does not fully meet PLOS ONE’s publication criteria as it currently stands. Therefore, we invite you to submit a revised version of the manuscript that addresses the points raised during the review process.

**ACADEMIC EDITOR: **

The paper needs a minor revision.

We look forward to receiving your revised manuscript.

Kind regards,

Faham Khamesipour, Ph.D.

Academic Editor

PLOS ONE

Journal Requirements:

Additional Editor Comments:

The paper needs a minor revision.

Reviewers' comments:

Reviewer's Responses to Questions

**Comments to the Author**

1. If the authors have adequately addressed your comments raised in a previous round of review and you feel that this manuscript is now acceptable for publication, you may indicate that here to bypass the “Comments to the Author” section, enter your conflict of interest statement in the “Confidential to Editor” section, and submit your "Accept" recommendation.

Reviewer #1: All comments have been addressed

Reviewer #2: (No Response)

2. Is the manuscript technically sound, and do the data support the conclusions?

Reviewer #1: Yes

Reviewer #2: Yes

3. Has the statistical analysis been performed appropriately and rigorously? 

Reviewer #1: Yes

Reviewer #2: Yes

4. Have the authors made all data underlying the findings in their manuscript fully available?

Reviewer #1: Yes

Reviewer #2: No

5. Is the manuscript presented in an intelligible fashion and written in standard English?

Reviewer #1: Yes

Reviewer #2: Yes

6. Review Comments to the Author

Reviewer #1: (No Response)

Reviewer #2: In the revised version, the authors provide more information and detail (esp. Supporting Information is valuable to understand the protocol), and the results are better documented. The conclusion has been much improved, with some suggestions how to improve the tool.

However, some of the author’s responses to reviewers’ comments raised eyebrows; it really underscores that the assessors were not properly trained and the results issued from the application of the tool are not standardized at all.

«It transpired that the assessors used the levels in a different way than instructed» Similarly, “Assessors were consulted, and they informed the authors that they considered both level one and 2 to imply that there was minimal risk for tail biting in the pen. Levels 3 and 4 were being applied interchangeably whenever a risk factor related to that risk category could be clearly identified”. Risk categories: “However these were inserted by an external party who was also involved in the development of the tool, without piloting the definitions with either potential assessors or other personnel involved in the tool development”

So assessors just decided upon themselves to use the levels interchangeably: these issues should have been reported by the assessors and the procedure adapted accordingly during the study.

It has to be made clear already in the abstract that the results have to be taken with a grain of salt, and that the tool lacks standardization and the assessors lack proper training. Readers should already know before they read the main text of the article that the authors tried to implement the tool but faced major issues and that they offer recommendations for those developing such tools in other countries. Knowing this beforehand avoids wasting readers’ time when they expect clear results.

“due to the poor performance of this risk assessment protocol, a second one was developed and now in phase of validation. The new risk assessment tool aims to be more specific, to help assessors (and farmers) to obtain a clear insight of weakness and strength of the farm.”

This raises the question if it would not have been better to wait for those results before publishing. There should be a clear benefit in publishing the failed tool for other researchers! Again, it should be highlighted in the abstract.

Overall, the results are still a bit difficult to understand and it is not clear what authors exactly want to show.

Table 7: I do not understand this table: if this is in a sample of pens where there was no risks then why is there a line with major risks.

Figure 6: in the method there are now 3 levels: 0 being when the risk was not observed, then low risk and high risk. Here I suggest changing 0 and 1 to low and high if that is what it refers to.

L 290: suggestion: random effect instead of repeated effect.

7. PLOS authors have the option to publish the peer review history of their article (what does this mean?). If published, this will include your full peer review and any attached files.

Reviewer #1: **Yes: **Ryan Samuel

Reviewer #2: No

---

## [Author Response · Author response to Decision Letter 1]

9 Apr 2024

In the revised version, the authors provide more information and detail (esp. Supporting Information is valuable to understand the protocol), and the results are better documented. The conclusion has been much improved, with some suggestions how to improve the tool.

AU: We thank the reviewer for their comments, and appreciation of how we have improved the paper

However, some of the author’s responses to reviewers’ comments raised eyebrows; it really underscores that the assessors were not properly trained and the results issued from the application of the tool are not standardized at all.

«It transpired that the assessors used the levels in a different way than instructed» Similarly, “Assessors were consulted, and they informed the authors that they considered both level one and 2 to imply that there was minimal risk for tail biting in the pen. Levels 3 and 4 were being applied interchangeably whenever a risk factor related to that risk category could be clearly identified”. Risk categories: “However these were inserted by an external party who was also involved in the development of the tool, without piloting the definitions with either potential assessors or other personnel involved in the tool development”

So assessors just decided upon themselves to use the levels interchangeably: these issues should have been reported by the assessors and the procedure adapted accordingly during the study.

AU: We apologise for the lack of clarity as to why the study was carried out in the way it was. The aim was to evaluate the first Irish national tail biting risk assessment scheme. The risk assessment tool that was used as part of the scheme was developed by the Department of Agriculture, Food and the Marine, to meet the requirement by the EU that all member states should put in place a system of risk assessment for tail biting on farm. Following the tool development, and without any on farm testing, a scheme was established so that producers could have the assessment performed free of charge by their PVP (who were paid for their time by funding delivered via Animal Health Ireland (AHI)). The scheme was also managed by AHI, an industry-led, not-for-profit partnership between livestock producers, processors, animal health advisers and government. AHI co-ordinated and delivered training for the assessors, and stored the results of the assessment tool for each farm in a database. The role of the researchers in this project was simply to assess the effectiveness of the scheme in assessing risk for tail biting, and if it wasn’t effective, to try to understand why. The reviewer is completely correct, in that what we found is that the training was not sufficient to ensure that all the assessors were using the system in a consistent manner. This is partly because of the way the tool was designed, with as it turns out a lack of clarity as to how to assign risk level. However it was beyond the remit of the research team who ran the study to change procedures, or the definitions in the tool, our role was simply to gather and compile data, and analyse and report it. It was also not clear until we obtained the audit reports and examined the data where the problems lay. We have made some changes to the abstract, introduction, and discussion so that this aim is clearer. 

It has to be made clear already in the abstract that the results have to be taken with a grain of salt, and that the tool lacks standardization and the assessors lack proper training. Readers should already know before they read the main text of the article that the authors tried to implement the tool but faced major issues and that they offer recommendations for those developing such tools in other countries. Knowing this beforehand avoids wasting readers’ time when they expect clear results.

AU: We have edited the text in the abstract in line with your suggestion.

“due to the poor performance of this risk assessment protocol, a second one was developed and now in phase of validation. The new risk assessment tool aims to be more specific, to help assessors (and farmers) to obtain a clear insight of weakness and strength of the farm.”

This raises the question if it would not have been better to wait for those results before publishing. There should be a clear benefit in publishing the failed tool for other researchers! Again, it should be highlighted in the abstract.

AU: The new tool is completely different, and has been developed from scratch, it is not an adaptation of the existing one, so the two are not comparable. We are planning to publish the results from our assessment of this new tool in a subsequent paper once the research is complete. We consider that the results from this study will be beneficial to other researchers, because it highlights ‘what not to do’, which is essential knowledge to help both researchers and policy makers in avoiding future mistakes. There have now been hundreds of assessments carried out using this tool by PVPs across Ireland, so a considerable amount of time and finances have been invested in the scheme, which is not providing the benefits that it aimed to.

Overall, the results are still a bit difficult to understand and it is not clear what authors exactly want to show.

AU: We understand that there is a lot of description, so we outlined the four aims of the analysis in the statistical analysis section (L277-281), and used the same headings in the results.

Table 7: I do not understand this table: if this is in a sample of pens where there was no risks then why is there a line with major risks.

AU: Assessors were asked to state whether they considered a pen was at risk or not overall (L135) as well as considering the level of risk that they considered each pen had in each of 6 risk categories (L136-142). It transpired that even when assessors considered that there was no overall risk, they sometimes still assigned levels of risk within the categories. Table 7 considers only the pens where the assessors considered there was no overall level of risk, and provides a breakdown of how they assigned the level of risk for each risk category. We have edited the text in this section (L385-390) to try and improve the explanation of what is in the table.

Figure 6: in the method there are now 3 levels: 0 being when the risk was not observed, then low risk and high risk. Here I suggest changing 0 and 1 to low and high if that is what it refers to.

AU: We apologise that the legend may have been misleading, we had updated it in the text, but not on the figure itself, which we have now done. For this, we only considered whether the assessors considered the pens at risk or not at risk overall (as per L135). We did not consider the level of risk assigned to each of the risk categories.

L 290: suggestion: random effect instead of repeated effect.

AU: We considered it a repeated effect because we were repeating the measurements within the experimental unit (the farm). However, repeated effects in linear mixed models are treated as random effects so they are computationally the same

---

## [Decision Letter · Decision Letter 2]

28 May 2024

PONE-D-23-38802R2Evaluation of a scheme to identify risks for tail biting in pigsPLOS ONE

Dear Dr. O'Driscoll,

Thank you for submitting your manuscript to PLOS ONE. After careful consideration, we feel that it has merit but does not fully meet PLOS ONE’s publication criteria as it currently stands. Therefore, we invite you to submit a revised version of the manuscript that addresses the points raised during the review process.

I would like to say that I think that the paper has improved and it is suitable for publication after some minor comments.

We look forward to receiving your revised manuscript.

Kind regards,

Faham Khamesipour, Ph.D.

Academic Editor

PLOS ONE

Journal Requirements:

Additional Editor Comments:

I would like to say that I think that the paper has improved and it is suitable for publication after some minor comments.

Comments for authors

Lines 256-260: methodological question. The risk on doing this pen level estimate is that some pigs had a high number of lesions. Do you have a measure of disperson (Standard error), or do you work with the median or some parameters which avoids not considering the variation in data? I am not sure if this is reflected in your results in Table 8, since the interquartile range you provide is already of pens not of individuals within a pen.

Another variable, ‘Visible injuries’ was created by summing the number of tail, ear, flank and aggression lesions in the pen, and dividing by the number of pigs per pen to provide a pen level estimate relative to the number of pigs

Line 291. It seems that pens were only observed once, why is it used as repeated effect?

Line 355. For the undocked pigs, it is understood that you used the same scoring system at the abattoir? Do you think you should use in futur studies a different one, which could evaluate lenght?

Lines 571-578. Take into account my previous comment on the fact that when you score at pen level injuries, one biased factor could be disperion (ie animals heavily injured increasing the overall score)

Line 614- 625. In my opinion, this problem about lack of correlation between on farm risk assessment and at abattoir final result could be worse in undocked pigs. Your sample size is probably not sufficient to separate both types of pigs, but it would be an interesting study for the study. Not only because tail biting is still more problemàtic in undocked pigs, but because the scoring system at the abattoir needs to be able to disentangle those pigs which has suffered tail biting but lesion has healed over time.

Reviewers' comments:

Reviewer's Responses to Questions

**Comments to the Author**

1. If the authors have adequately addressed your comments raised in a previous round of review and you feel that this manuscript is now acceptable for publication, you may indicate that here to bypass the “Comments to the Author” section, enter your conflict of interest statement in the “Confidential to Editor” section, and submit your "Accept" recommendation.

Reviewer #5: All comments have been addressed

2. Is the manuscript technically sound, and do the data support the conclusions?

Reviewer #5: Yes

3. Has the statistical analysis been performed appropriately and rigorously? 

Reviewer #5: Yes

4. Have the authors made all data underlying the findings in their manuscript fully available?

Reviewer #5: (No Response)

5. Is the manuscript presented in an intelligible fashion and written in standard English?

Reviewer #5: Yes

6. Review Comments to the Author

Reviewer #5: (No Response)

7. PLOS authors have the option to publish the peer review history of their article (what does this mean?). If published, this will include your full peer review and any attached files.

Reviewer #5: **Yes: **Emma Fàbrega Romans

---

## [Author Response · Author response to Decision Letter 2]

31 May 2024

Lines 256-260: methodological question. The risk on doing this pen level estimate is that some pigs had a high number of lesions. Do you have a measure of disperson (Standard error), or do you work with the median or some parameters which avoids not considering the variation in data? I am not sure if this is reflected in your results in Table 8, since the interquartile range you provide is already of pens not of individuals within a pen.

Another variable, ‘Visible injuries’ was created by summing the number of tail, ear, flank and aggression lesions in the pen, and dividing by the number of pigs per pen to provide a pen level estimate relative to the number of pigs

AU: We agree with the problem you have identified, thank you for this important observation. It is possible that either a small number of pigs had a lot of lesions, or many pigs had a low number, and there is no distinction between which is the case. However, unfortunately the data that the assessors collected did not go into this level of detail, all that was on the report was the total number of lesions that they saw in the pen, so it was not possible to calculate the median lesion number per pig. Because all we could do was calculate the mean (no. lesions/no. pigs / pen) this is the value we used in the analysis, which was carried out using a non-parametric test. We added some text from L572-576 to acknowledge this issue.

Line 291. It seems that pens were only observed once, why is it used as repeated effect?

AU: Yes, each pen was only observed once, but we considered the farm the experimental unit, so each pen was a repeated measure within each farm

Line 355. For the undocked pigs, it is understood that you used the same scoring system at the abattoir? Do you think you should use in futur studies a different one, which could evaluate lenght?

AU: This refers to L255 I think. There was only one scoring system because only a tiny proportion of pigs in Ireland are not docked. In the study population it was less than 1%. However yes, we agree that in future on populations of pigs where the tails are not docked a measure of length would be very useful!

Lines 571-578. Take into account my previous comment on the fact that when you score at pen level injuries, one biased factor could be disperion (ie animals heavily injured increasing the overall score)

AU: See answer above

Line 614- 625. In my opinion, this problem about lack of correlation between on farm risk assessment and at abattoir final result could be worse in undocked pigs. Your sample size is probably not sufficient to separate both types of pigs, but it would be an interesting study for the study. Not only because tail biting is still more problemàtic in undocked pigs, but because the scoring system at the abattoir needs to be able to disentangle those pigs which has suffered tail biting but lesion has healed over time.

AU: We agree, this would be in interesting research question for a future project. However at the moment there are not sufficient pigs in Ireland where tails are not docked to carry out this evaluation. We have added some text to L640-643

---

## [Decision Letter · Decision Letter 3]

10 Jun 2024

Evaluation of a scheme to identify risks for tail biting in pigs

PONE-D-23-38802R3

Dear Dr. O'Driscoll,

We’re pleased to inform you that your manuscript has been judged scientifically suitable for publication and will be formally accepted for publication once it meets all outstanding technical requirements.

Kind regards,

Faham Khamesipour, Ph.D.

Academic Editor

PLOS ONE

Additional Editor Comments (optional):

The comments the reviewers made have been addressed. I think this kind of practical information is very useful when published for researchers to know what happens in commercial settings and address future research topics.

Reviewers' comments:

Reviewer's Responses to Questions

**Comments to the Author**

1. If the authors have adequately addressed your comments raised in a previous round of review and you feel that this manuscript is now acceptable for publication, you may indicate that here to bypass the “Comments to the Author” section, enter your conflict of interest statement in the “Confidential to Editor” section, and submit your "Accept" recommendation.

Reviewer #5: All comments have been addressed

2. Is the manuscript technically sound, and do the data support the conclusions?

Reviewer #5: Yes

3. Has the statistical analysis been performed appropriately and rigorously? 

Reviewer #5: Yes

4. Have the authors made all data underlying the findings in their manuscript fully available?

Reviewer #5: Yes

5. Is the manuscript presented in an intelligible fashion and written in standard English?

Reviewer #5: Yes

6. Review Comments to the Author

Reviewer #5: Dear authors

Many thanks for considering my comments. The paper is now suitable for publication. Nice piece of work.

7. PLOS authors have the option to publish the peer review history of their article (what does this mean?). If published, this will include your full peer review and any attached files.

Reviewer #5: **Yes: **Emma Fabrega Romans

---

## [Editor Report · Acceptance letter]

19 Jun 2024

PONE-D-23-38802R3 

PLOS ONE

Dear Dr. O'Driscoll, 

I'm pleased to inform you that your manuscript has been deemed suitable for publication in PLOS ONE. Congratulations! Your manuscript is now being handed over to our production team.

Kind regards, 

on behalf of

Dr. Faham Khamesipour 

Academic Editor

PLOS ONE